# AutoSKDBERT: Learn to Stochastically Distill BERT

## Abstract

In this paper, we propose AutoSKDBERT, a new knowledge distillation paradigm for BERT compression, that stochastically samples a teacher from a predefined teacher team following a categorical distribution in each step, to transfer knowledge into student. AutoSKDBERT aims to discover the optimal categorical distribution which plays an important role to achieve high performance. The optimization procedure of AutoSKDBERT can be divided into two phases: 1) phase-1 optimization distinguishes effective teachers from ineffective teachers, and 2) phase-2 optimization further optimizes the sampling weights of the effective teachers to obtain satisfactory categorical distribution. Moreover, after phase-1 optimization completion, AutoSKDBERT adopts teacher selection strategy to discard the ineffective teachers whose sampling weights are assigned to the effective teachers. Particularly, to alleviate the gap between categorical distribution optimization and evaluation, we also propose a stochastic single-weight optimization strategy which only updates the weight of the sampled teacher in each step. Extensive experiments on GLUE benchmark show that the proposed AutoSKDBERT achieves state-of-the-art score compared to previous compression approaches on several downstream tasks, including pushing MRPC F1 and accuracy to 93.2 (0.6 point absolute improvement) and 90.7 (1.2 point absolute improvement), RTE accuracy to 76.9 (2.9 point absolute improvement).

## 1 Introduction

BERT (Devlin et al., 2019) has brought about a sea change in the field of Natural Language Processing (NLP). Following BERT, numerous subsequent works focus on various perspectives to further improve its performance, e.g., hyper-parameter (Liu et al., 2019b), pre-training corpus (Liu et al., 2019b; Raffel et al., 2020), learnable embedding paradigm (Raffel et al., 2020), pre-training task (Clark et al., 2020), architecture (Gao et al., 2022) and self-attention (Shi et al., 2021), etc. However, there are massive redundancies in the above BERT-style models w.r.t. attention heads (Michel et al., 2019; Dong et al., 2021), weights (Gordon et al., 2020), and layers (Fan et al., 2020). Consequently, many compact BERT-style language models are proposed via pruning (Fan et al., 2020; Guo et al., 2019), quantization (Shen et al., 2020), parameter sharing (Lan et al., 2020) and Knowledge Distillation (KD) (Iandola et al., 2020; Pan et al., 2021). In this paper, we focus on the KD-based compression approaches.

From the point of view of learning procedure, KD is used in the pre-training (Turc et al., 2019; Sanh et al., 2019; Sun et al., 2020; Jiao et al., 2020) and fine-tuning phases (Sun et al., 2019; Jiao et al., 2020; Wu et al., 2021). On the other hand, from the point of view of distillation objective, KD is employed for the outputs of hidden layer (Sun et al., 2020), final layer (Wu et al., 2021), embedding (Sanh et al., 2019) and self-attention (Wang et al., 2020). Wu et al. (2021) employ multiple teachers to achieve better performance than single-teacher KD based approaches on several downstream tasks of GLUE benchmark (Wang et al., 2019). As shown in Table 1, nevertheless, the ensemble of multiple teachers are not always more effective than the single teacher for student distillation. There are two possible reasons: 1) *diversity losing* (Tran et al., 2020) and 2) *capacity gap* (Mirzadeh et al., 2020). On the one hand, the ensemble prediction of multi-teacher KD loses the diversity of each teacher. On the other hand, between the large-capacity teacher ensemble and small-capacity student, there is a capacity gap which can be prone to unsatisfactory distillation performance.

Table 1: Performances of knowledge distillation using single and multiple teachers for a 6-layer BERT-style language model on the development set of GLUE benchmark. In this experiment, we employ five teachers, i.e. $T_{10}$ to $T_{14}$ shown in Appendix C.1, for single-teacher distillation and multi-teacher distillation. We introduce the implementation details in Appendix H.

| Task | MRPC | RTE | CoLA | SST-2 | QQP | QNLI | MNLI |
|---|---|---|---|---|---|---|---|
| Metrics | $\frac{F1+acc}{2}$ | acc | Mcc | acc | $\frac{F1+acc}{2}$ | acc | m |
| Best Single Teacher† | $T_{13}$ | $T_{10}$ | $T_{10}$ | $T_{12}$ | $T_{11}$ | $T_{12}$ | $T_{14}$‡ |
| Single-teacher KD | **90.0** | 73.3 | 49.3 | **93.1** | **89.0** | **91.4** | 83.5 |
| Multi-teacher KD | 89.7 | **73.7** | **50.1** | 92.2 | 88.6 | 91.1 | **83.6** |
| Gain | -0.3 | +0.4 | +0.8 | -0.9 | -0.4 | -0.3 | +0.1 |

† The best teacher for student distillation on each downstream task as shown in Table 15.
‡ Pre-training with whole word masking.

To solve the above mentioned issues, we propose AutoSKDBERT which stochastically samples a teacher from a predefined teacher team following a categorical distribution in each step, to transfer knowledge into student. The task of AutoSKDBERT is learning the optimal categorical distribution to achieve high performance. 1) Given a teacher team which consists of multiple teachers with multi-level capacities, AutoSKDBERT optimizes an initialized categorical distribution to distinguish effective teachers from ineffective teachers in phase-1 optimization. 2) The sampling weights of the ineffective teachers are assigned to the effective teachers via teacher selection strategy after phase-1 optimization completion. 3) AutoSKDBERT further optimizes the weights of the effective teachers rather than the ineffective teachers' in phase-2 optimization. We implement extensive experiments on GLUE benchmark (Wang et al., 2019) to verify the effectiveness of the proposed AutoSKD-BERT. Moreover, to show the generalization capacity, we have also distilled deep convolutional neural network (e.g., ResNet (He et al., 2016), Wide ResNet (Zagoruyko & Komodakis, 2016)) by AutoSKDBERT for image classification on CIFAR-100 (Krizhevsky et al., 2009), as shown in Appendix B. Our contributions are summarized as follows[1]:

- We propose AutoSKDBERT which stochastically samples a teacher from the predefined teacher team following the categorical distribution in each step, to transfer knowledge into the student of BERT-style language model.

- We propose a two-phase optimization framework with teacher selection strategy to select effective teachers and learn the optimal categorical distribution in a differentiable way.

- We propose Stochastic Single-Weight Optimization (SSWO) strategy to alleviate the consistency gap between the categorical distribution optimization and evaluation for performance improvement.

## 2 THE PROPOSED AUTOSKDBERT

### 2.1 OVERVIEW

In each step, AutoSKDBERT samples a teacher $\hat{T}$ from a teacher team which consists of $n$ multi-level BERT-style teachers $T_{1:n}$, to transfer knowledge into student S. The objective function of AutoSKDBERT can be expressed as

$$\mathcal{L}(w) = \sum_{x \in \mathcal{X}} \mathcal{L}_d(f_{\hat{T} \in T_{1:n}}(x), f_S(x; w)), \quad (1)$$

where $\mathcal{L}_d$ represents distilled loss function to compute the difference between the student S with learnable parameter $w$ and the sampled teacher $\hat{T}$, $\mathcal{X}$ denotes the training data, $f_{\hat{T} \in T_{1:n}}(\cdot)$ and $f_S(\cdot)$ denote the logits from $\hat{T}$ and S, respectively.

In AutoSKDBERT, a categorical distribution $\text{Cat}(\theta)$ where $\theta = \{\theta_{1:n}\}$ and $\sum_{i=1}^{n} \theta_i = 1$, is employed to sample the teacher from the teacher team. Particularly, the probability $p(T_i)$ of $T_i$ being sampled is $\theta_i$. We observe that $\text{Cat}(\theta)$ plays an important role for obtaining high performance of AutoSKDBERT. As a result, the task of AutoSKDBERT then turns into learning the optimal categorical distribution $\text{Cat}(\theta^*)$, as illustrated in Figure 1.

---

[1]The code will be made publicly available upon publication of the paper.

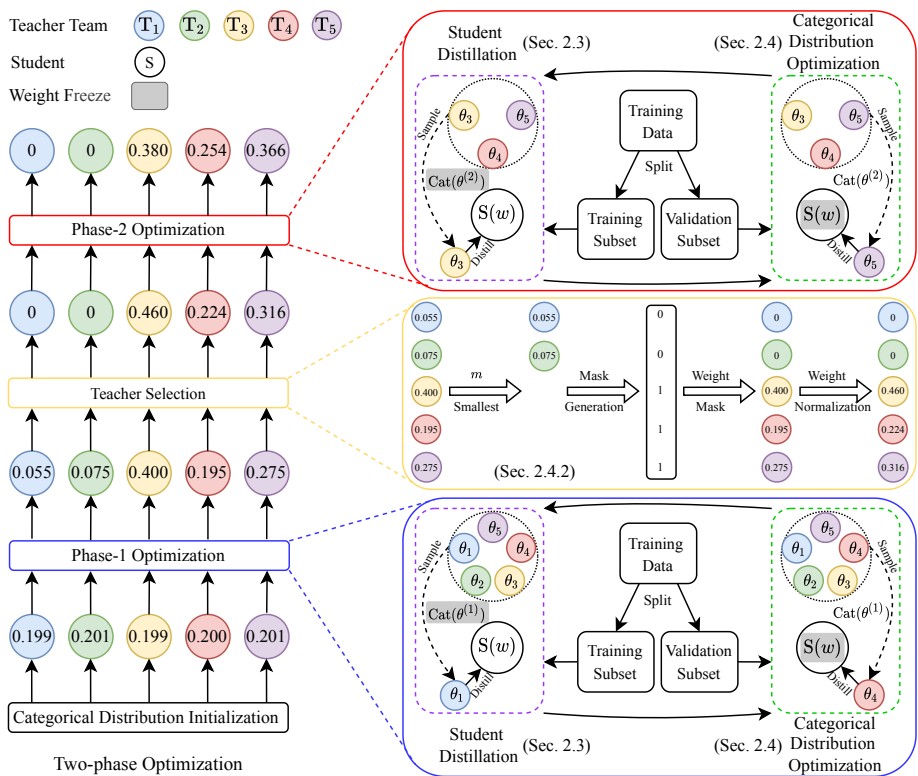

Figure 1: Two-phase optimization framework with teacher selection strategy for AutoSKDBERT. 1) For a pre-defined teacher team, AutoSKDBERT optimizes an initialized categorical distribution to distinguish effective teachers from ineffective teachers. 2) After phase-1 optimization completion, the sampling weights of the ineffective teachers are assigned to the effective teachers via teacher selection strategy. 3) AutoSKDBERT further optimizes the weights of the effective teachers rather than the ineffective teachers in phase-2 optimization. Best viewed in color.

## 2.2 PROBLEM FORMULATION

AutoSKDBERT has two groups of learnable parameter: 1) $w$ of student and 2) $\theta$ of categorical distribution. We split original training data into training and validation subsets, and denote $\mathcal{L}_{train}$ and $\mathcal{L}_{val}$ as the losses on training and validation subsets, respectively. Both $\mathcal{L}_{train}$ and $\mathcal{L}_{val}$ are determined not only by $\mathrm{Cat}(\theta)$, but also by $w$. Particularly, AutoSKDBERT aims to learn the best categorical distribution $\mathrm{Cat}(\theta^*)$ that minimizes the validation loss $\mathcal{L}_{val}(w^*, \mathrm{Cat}(\theta))$, where the weights $w^*$ associated with the categorical distribution $\mathrm{Cat}(\theta)$ are obtained by $\mathrm{argmin}_w \ \mathcal{L}_{train}(w, \mathrm{Cat}(\theta))$. Consequently, AutoSKDBERT can be considered as a bilevel optimization problem (Colson et al., 2007) with upper-level variable $\mathrm{Cat}(\theta)$ and lower-level variable $w$:

$$\min_{\mathrm{Cat}(\theta)} \quad \mathcal{L}_{val}(w^*(\mathrm{Cat}(\theta)), \mathrm{Cat}(\theta)),$$
$$\text{s.t.} \quad w^*(\mathrm{Cat}(\theta)) = \underset{w}{\mathrm{argmin}} \ \mathcal{L}_{train}(w, \mathrm{Cat}(\theta)). \tag{2}$$

We optimize $w$ of student (see Section 2.3) and $\theta$ of categorical distribution (see Section 2.4) in an alternate and iterative way, and show the optimization algorithm in Algorithm 1.

## 2.3 STUDENT DISTILLATION

For student distillation, $\mathrm{Cat}(\theta)$ is frozen. Similar to Eq. 1, we utilize the following object function:

$$\mathcal{L}(w) = \sum_{x \in \mathcal{X}} \mathcal{L}_d(\hat{\theta} f_{\hat{\mathrm{T}} \in \mathrm{T}_{1:n}}(x), f_{\mathrm{S}}(x; w)), \tag{3}$$

where $\hat{\theta}$ indicates the probability of the teacher $\hat{\mathrm{T}}$ being sampled from $\mathrm{T}_{1:n}$ according to $\mathrm{Cat}(\theta)$.

---

**Algorithm 1:** Two-phase Optimization for AutoSKDBERT

---

Initialize categorical distribution $\text{Cat}(\theta^{(1)})$ for phase-1 optimization, weights $w$ of student,
  maximum step $N$, current step $n = 0$;
**while** $n < \frac{N}{2}$ **do**

 Update $\text{Cat}(\theta^{(1)})$ by descending Eq. 7 ;                  // phase-1 categorical
  distribution optimization
 Update $w$ by descending Eq. 3 ;          // phase-1 student distillation
 $n = n + 1$;

**end**
Select effective teachers to generate $\text{Cat}(\theta^{(2)})$ by Eq. 8;          // teacher selection
**while** $\frac{N}{2} \leq n < N$ **do**

 Update $\text{Cat}(\theta^{(2)})$ by descending Eq. 7 ;                  // phase-2 categorical
  distribution optimization
 Update $w$ by descending Eq. 3 ;          // phase-2 student distillation
 $n = n + 1$;

**end**

---

### 2.4 CATEGORICAL DISTRIBUTION OPTIMIZATION

For categorical distribution optimization, $w$ is frozen. We propose a two-phase optimization framework with teacher selection strategy to learn appropriate categorical distribution:

1. **Phase-1 Optimization** distinguishes effective teachers from ineffective teachers in the teacher team according to $\text{Cat}(\theta)$;

2. **Teacher Selection** discards the ineffective teachers whose weights are assigned to the effective teachers;

3. **Phase-2 Optimization** further optimizes the weights of the effective teachers rather than the ineffective teachers;

where a Stochastic Single-Weight Optimization (SSWO) strategy is proposed for categorical distribution optimization. Below, categorical distribution optimization and teacher selection strategy are introduced in detail.

### 2.4.1 CATEGORICAL DISTRIBUTION OPTIMIZATION VIA SSWO

To optimize $\text{Cat}(\theta)$ in a differentiable way, Continuous Relaxation (CR) (Liu et al., 2019a) is a common technique to obtain mixture of logits w.r.t. teachers as

$$\overline{f}_{\text{T}_{1:n}}(x; \text{Cat}(\theta)) = \sum_{i=1}^{n} \theta_i f_{\text{T}_i}(x). \tag{4}$$

Subsequently, $\text{Cat}(\theta)$ can be optimized by an approximation scheme:

$$\nabla_{\text{Cat}(\theta)} \mathcal{L}_{val}(w^*(\text{Cat}(\theta)), \text{Cat}(\theta)) \approx \nabla_{\text{Cat}(\theta)} \mathcal{L}_{val}(w - \alpha \nabla_w \mathcal{L}_{train}(w, \text{Cat}(\theta)), \text{Cat}(\theta)), \tag{5}$$

where $w$ and $\alpha$ indicate the current weights of the student and the learning rate of categorical distribution, respectively. In particular, we employ $w$ with a single-step adapting (i.e., $w - \alpha \nabla_w \mathcal{L}_{train}(w, \text{Cat}(\theta))$ to appropriate $w^*(\text{Cat}(\theta))$ for avoiding the inner optimization in Eq. 2. This appropriation scheme has been widely used in meta-learning (Finn et al., 2017) and neural architecture search (Liu et al., 2019a).

However, in the case of CR, there is a consistency gap between the categorical distribution optimization and evaluation in terms of the teacher's logits. For categorical distribution optimization, $\overline{f}_{\text{T}_{1:n}}(x; \text{Cat}(\theta))$ is used to compute the difference between the student's logits as $\sum_{x \in \mathcal{X}} \mathcal{L}_d(\overline{f}_{\text{T}_{1:n}}(x; \text{Cat}(\theta)), f_{\text{S}}(x; w))$. For categorical distribution evaluation, however, only the logits of the sampled teacher $f_{\hat{\text{T}} \in \text{T}_{1:n}}(x)$ is used to obtain the difference to the student's logits as $\sum_{x \in \mathcal{X}} \mathcal{L}_d(f_{\hat{\text{T}} \in \text{T}_{1:n}}(x), f_{\text{S}}(x; w))$.

To alleviate the consistency gap, we propose SSWO whose objective function can be written as

$$\mathcal{L}(w; \hat{\theta}) = \sum_{x \in \mathcal{X}} \mathcal{L}_d(\hat{\theta} f_{\hat{\mathrm{T}} \in \mathrm{T}_{1:n}}(x), f_{\mathrm{S}}(x; w)), \qquad (6)$$

where $\hat{\theta}$ plays also a role like label smoothing (Szegedy et al., 2016) which aims to reduce the confidence coefficient of the sampled teacher and avoid over fitting (Müller et al., 2019) of the categorical distribution. Moreover, the smaller the sampling weight, the more reduction the confidence coefficient of the sampled teacher. Subsequently, the sampled single-weight $\hat{\theta}$ can be optimized by

$$\nabla_{\hat{\theta} \sim \mathrm{Cat}(\theta)} \mathcal{L}_{val}(w^*(\hat{\theta}), \hat{\theta}) \approx \nabla_{\hat{\theta} \sim \mathrm{Cat}(\theta)} \mathcal{L}_{val}(w - \alpha \nabla_w \mathcal{L}_{train}(w, \hat{\theta}), \hat{\theta}). \qquad (7)$$

In practice, the proposed SSWO achieves better performance than CR, as shown in Section 4.2.

### 2.4.2 TEACHER SELECTION

After phase-1 optimization completion, $m$ ineffective teachers are separated from the teacher team according to the current categorical distribution $\mathrm{Cat}(\theta^{(1)})$, where the smaller the weight, the more ineffective the teacher. For avoiding categorical distribution optimizing from scratch, we present teacher selection strategy which assigns the weights of $m$ ineffective teachers to $n - m$ effective teachers, to deliver the categorical distribution $\mathrm{Cat}(\theta^{(2)})$ for phase-2 optimization by

$$\mathrm{Cat}(\theta^{(2)}) = \frac{\mathrm{Cat}(\theta^{(1)}) mask(m\_smallest(\mathrm{Cat}(\theta^{(1)}), m))}{\max(\|\mathrm{Cat}(\theta^{(1)}) mask(m\_smallest(\mathrm{Cat}(\theta^{(1)}), m))\|_p, \epsilon)}, \qquad (8)$$

where $p$ (1 in this paper) denotes the exponent value in the norm formulation, $\epsilon$ is a small value (1e-12 in this paper) to avoid division by zero, $m\_smallest(\mathrm{Cat}(\theta^{(1)}), m)$ obtains $m$ indexes of ineffective teachers according to $\mathrm{Cat}(\theta^{(1)})$, and $mask(\cdot)$ generates a mask where the values of $m$ ineffective and $n - m$ effective teachers are set to 0 and 1, respectively.

## 3 EXPERIMENTS AND RESULTS

### 3.1 DATASETS AND SETTINGS

**Datasets.** We evaluate the proposed AutoSKDBERT on GLUE benchmark (Wang et al., 2019), including MRPC (Dolan & Brockett, 2005), RTE (Bentivogli et al., 2009), CoLA (Warstadt et al., 2019), SST-2 (Socher et al., 2013), QQP (Chen et al., 2018), QNLI (Rajpurkar et al., 2016) and MNLI (Williams et al., 2017). Moreover, STS-B (Cer et al., 2017) is not selected.

**Settings.** We employ the development set of GLUE benchmark dubbed as GLUE-dev, for categorical distribution evaluation of AutoSKDBERT. We employ a teacher team which consists of 14 BERT-style teachers, to distill a 6-layer BERT-style student dubbed AutoSKDBERT. The architecture information of the student and the teachers can be found in Appendix C.1. On the one hand, we employ weak $\mathrm{T}_{01}$ to $\mathrm{T}_{09}$ (refer to Table 12) to verify a guess that the diversities of those weak teachers contribute to improve the distillation performance or not. On the other hand, under a conclusion that the extreme strong teacher (i.e., $\mathrm{T}_{13}$ and $\mathrm{T}_{14}$) can not always contribute to improving the distillation performance (see Appendix B.4 and F in the revised manuscript), we employ strong $\mathrm{T}_{13}$ and $\mathrm{T}_{14}$ to verify the effectiveness of the proposed distillation paradigm for capacity gap alleviation. We give a general way to design the teacher team and determine the value of $m$ in Appendix A.

### 3.2 TWO-PHASE OPTIMIZATION

We employ identical experimental settings for student distillation and categorical distribution optimization in both phase-1 and phase-2 optimization. The original training set is split fifty-fifty into two subsets, i.e., training subset for student distillation (see Section 2.3) and validation subset for categorical distribution optimization (see Section 2.4).

### 3.2.1 STUDENT DISTILLATION

We choose Adam with a weight decay of 1e-4 as the optimizer for student distillation. For various downstream tasks, we employ different batch size, learning rate and epoch number as shown in Table 2. Other hyper-parameters can be found in Appendix C.2.

Table 2: The hyper-parameters for student distillation.

| Hyper-parameter | MRPC | RTE | CoLA | SST-2 | QQP | QNLI | MNLI |
|---|---|---|---|---|---|---|---|
| Batch Size | 32 | 32 | 16 | 64 | 32 | 32 | 32 |
| Learning Rate | 1e-5 | 1e-5 | 1e-5 | 1e-5 | 3e-5 | 2e-5 | 2e-5 |
| Epoch Number | 50 | 50 | 50 | 10 | 2 | 10 | 2 |

### 3.2.2 CATEGORICAL DISTRIBUTION OPTIMIZATION

For categorical distribution optimization, we employ other Adam with a weight decay of 1e-3 as the optimizer. There are two important hyper-parameters: 1) the number of the ineffective teacher and 2) learning rate for categorical distribution optimization. Similarly, for different downstream tasks, the above two parameters are various as shown in Table 9. Other hyper-parameters are identical to student distillation. The impact of each hyper-parameter is discussed in Appendix E.

Table 3: The hyper-parameters for categorical distribution optimization.

| Hyper-parameter | MRPC | RTE | CoLA | SST-2 | QQP | QNLI | MNLI |
|---|---|---|---|---|---|---|---|
| Ineffective Teacher Number | 1 | 9 | 4 | 10 | 8 | 1 | 9 |
| Learning Rate | 9e-4 | 1e-3 | 7e-4 | 9e-4 | 1e-3 | 6e-4 | 4e-4 |

### 3.3 CATEGORICAL DISTRIBUTION EVALUATION

#### 3.3.1 IMPLEMENTATION DETAILS

AutoSKDBERT delivers 25 categorical distribution candidates in phase-2 optimization, and trains the student with 25 candidates from scratch to choose the optimal categorical distribution. In addition to epoch number, other hyper-parameters (e.g., batch size, learning rate, etc.) are identical to student distillation on various downstream tasks as shown in Table 2. The epoch number is set to 15 on MRPC, RTE, CoLA tasks, and 5 on SST-2, QQP, QNLI and MNLI tasks.

#### 3.3.2 LEARNED CATEGORICAL DISTRIBUTION

We show the categorical distributions learned by AutoSKDBERT on GLUE benchmark in Figure 2. Each teacher model shows various importances on different downstream tasks. 1) The strongest teacher $T_{14}$ plays a dominant role on CoLA, SST-2 and QNLI tasks. 2) Low-capacity teachers, e.g., $T_{02}$ to $T_{06}$, can also provide useful knowledge for student distillation on MRPC, CoLA and QNLI tasks. 3) The capacity of the effective teacher is not always larger than the discard teachers on RTE, SST-2 and QQP tasks. Moreover, the search and evaluation costs with respect to each downstream task are shown in Appendix F.

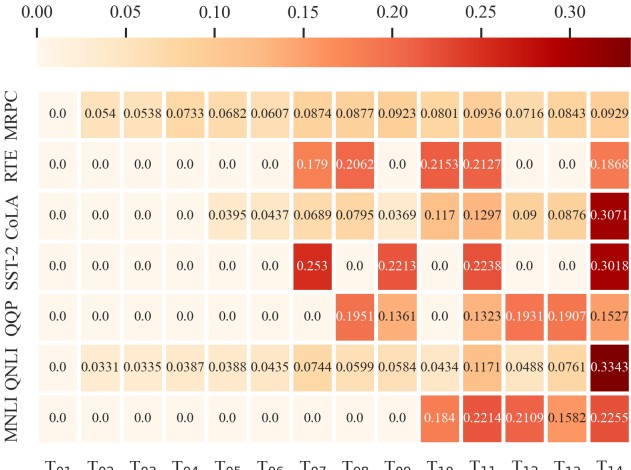

Figure 2: Categorical distributions learned on GLUE benchmark.

### 3.4 RESULTS AND ANALYSIS ON GLUE BENCHMARK

Table 4 summarizes the performance of AutoSKDBERT and the comparative approaches on GLUE-dev. The proposed AutoSKDBERT achieves state-of-the-art performance on four out of seven tasks. AutoSKDBERT contributes to achieving better performance on those tasks with small data size, e.g., MRPC and RTE. On MRPC, AutoSKDBERT achieves 93.2 F1 score and 90.7 accuracy score which are 0.6 and 1.2 point higher than previous state-of-the-art MoEBERT (Zuo et al., 2022), respectively.

Table 4: Results of AutoSKDBERT and other popular approaches on GLUE-dev. All comparative approaches have identical architecture, i.e., 6-layer BERT-style language model with 66 million parameters. † and ‡ indicate that the results are cited from Xu et al. (2020) and Zuo et al. (2022), respectively. ∗ means that the comparison between TinyBERT$_6$ and AutoSKDBERT may not be fair since the former employs GloVe word embedding (Pennington et al., 2014) based data augmentation and transformer-layer distillation and embedding-layer distillation. § indicates that the result is obtained by our settings with the distillation loss described in Wu et al. (2021), and the experimental details can be found in Appendix H. Moreover, the stronger teacher can not always contribute to improving the distillation performance of other approaches due to the capacity gap (Mirzadeh et al., 2020) as shown in Appendix G. Besides, we show also the performances of multi-teacher AvgKD and TAKD with $T_{01}$ to $T_{14}$ in Appendix C.4.

| Model | MRPC | RTE | CoLA | SST-2 | QQP | QNLI | MNLI |
|---|---|---|---|---|---|---|---|
| | F1/acc | acc | Mcc | acc | F1/acc | acc | m |
| Poor Man's BERT$_6$ (Sajjad et al., 2020) | -/80.2 | 65.0 | - | 90.3 | -/90.4 | 87.6 | 81.1 |
| DistilBERT$_6$ (Sanh et al., 2019) | 87.5/- | 59.9 | 51.3 | 92.7 | -/88.5 | 89.2 | 82.2 |
| LayerDrop (Fan et al., 2020)† | 85.9/- | 65.2 | 45.4 | 90.7 | -/88.3 | 88.4 | 80.7 |
| BERT-PKD (Sun et al., 2019)† | 85.7/- | 66.5 | 45.5 | 91.3 | -/88.4 | 88.4 | 81.3 |
| BERT-of-Theseus (Xu et al., 2020) | 89.0/- | 68.2 | 51.1 | 91.5 | -/89.6 | 89.5 | 82.3 |
| MiniLMv1 (Wang et al., 2020) | 88.4/- | 71.5 | 49.2 | 92.0 | -/91.0 | 91.0 | 84.0 |
| MiniLMv2 (Wang et al., 2020)‡ | 88.9/- | 72.1 | 52.5 | 92.4 | -/91.1 | 90.8 | 84.2 |
| TinyBERT$_6$ (Jiao et al., 2020)∗ | 90.6/89.3 | 73.4 | 54.0 | **93.0** | 88.0/91.1 | 91.1 | **84.5** |
| TinyBERT$_6$ (w/o aug) (Jiao et al., 2020)‡ | 88.4/- | 72.2 | 42.8 | 91.6 | -/90.6 | 90.5 | 83.5 |
| MT-BERT (Wu et al., 2021)§ | 90.8/87.0 | 72.2 | 49.1 | 92.2 | 87.1/90.4 | 91.4 | 83.8 |
| MoEBERT (Zuo et al., 2022) | 92.6/89.5 | 74.0 | **55.4** | **93.0** | **88.4/91.4** | 91.3 | **84.5** |
| AutoSKDBERT (Ours) | **93.2/90.7** | **76.9** | 51.8 | **93.0** | 88.0/91.0 | **91.6** | 84.3 |

On the other hand, compared to TinyBERT (Jiao et al., 2020) and MoEBERT (Zuo et al., 2022) on RTE task, AutoSKDBERT achieves 3.5 and 2.9 point absolute improvement, respectively.

However, on CoLA, TinyBERT and MoEBERT achieve 2.2 and 3.6 point absolute improvement compared to AutoSKDBERT, respectively. On the one hand, TinyBERT employs data augmentation and transformer layer distillation to achieve high performance. On the other hand, MoEBERT employs 1) more complex student whose architecture is an ensemble of multiple experts, and 2) extra distillation procedure, i.e., transformer layer distillation, to achieve novel performance.

The proposed approach is a general KD paradigm for BERT compression. Consequently, we implement also extensive experiments to verify the effectiveness for image classification on CIFAR-100 (see Appendix B) and the orthogonality with other approaches (see Appendix D).

## 4 ABLATION STUDIES

### 4.1 TWO-PHASE OPTIMIZATION: PHASE-1 VERSUS PHASE-2

In this section, AutoSKDBERT delivers also 25 categorical distribution candidates in phase-1 optimization. Subsequently, each categorical distribution candidate is trained from scratch using identical settings described in Section 3.3, and the best-performing one on each task is shown in Table 5.

Table 5: The performance of AutoSKDBERT with the best categorical distribution learned in phase-1 and phase-2 optimization on GLUE-dev.

| Task | MRPC | RTE | CoLA | SST-2 | QQP | QNLI | MNLI |
|---|---|---|---|---|---|---|---|
| **Metrics** | F1/acc | acc | Mcc | acc | F1/acc | acc | m |
| **Phase-1** | 92.9/90.2 | 73.7 | 49.2 | 92.9 | 87.6/90.7 | 91.3 | 83.0 |
| **Phase-2** | **93.2/90.7** | **76.9** | **51.8** | **93.0** | **88.0/91.0** | **91.6** | **84.3** |
| **Gain** | +0.3/+0.5 | +1.4 | +2.7 | +0.1 | +0.4/+0.3 | +0.3 | +1.3 |

Phase-2 optimization achieves better performance than phase-1 optimization, e.g., the absolute improvement is more than 1.3 on RTE, CoLA and MNLI, where those teachers weaker than the student are prone to providing useless knowledge even noise disturbance. However, low-capacity teachers contribute to improving the performance of AutoSKDBERT on MRPC, SST-2 and QNLI.

## 4.2 CATEGORICAL DISTRIBUTION UPDATE STRATEGY: CR-BASED VERSUS SSWO-BASED

In Section 2.4.1, we propose SSWO which stochastically samples a single-weight to optimize the categorical distribution, to alleviate the consistency gap between the categorical distribution optimization and evaluation of CR in terms of teachers' logits. For AutoSKDBERT with CR, the used hyper-parameters of categorical distribution optimization and evaluation are identical to AutoSKD-BERT with SSWO, as described in Section 3.2 and Section 3.3.

The proposed SSWO achieves better performance than CR on all tasks as shown in Table 6. Particularly, the absolute improvement is more than 1.6 point on RTE and CoLA tasks. Compared to Table 5, AutoSKDBERT with CR achieves higher performance than phase-1 AutoSKDBERT on six out of seven tasks. Consequently, useless teachers lead to more performance degradation than the consistency gap issue.

Table 6: The performance of AutoSKDBERT with the best categorical distribution learned by CR and SSWO for categorical distribution optimization on GLUE-dev.

| Task | MRPC | RTE | CoLA | SST-2 | QQP | QNLI | MNLI |
|---|---|---|---|---|---|---|---|
| Metrics | F1/acc | acc | Mcc | acc | F1/acc | acc | m |
| CR | 92.9/90.2 | 74.7 | 50.2 | 92.7 | 87.9/**91.0** | 91.5 | 83.9 |
| SSWO | **93.2/90.7** | **76.9** | **51.8** | **93.0** | **88.0/91.0** | **91.6** | **84.3** |

In addition to SSWO, heuristic optimization algorithms like evolutionary algorithm and reinforcement learning can also be used to determine the categorical distribution. In this paper, we choose the most efficient one, i.e., gradient-based SSWO.

## 4.3 CATEGORICAL DISTRIBUTION GENERATION: RANDOM VERSUS LEARNING

We compare two groups of implementation of AutoSKDBERT with various algorithms for categorical distribution generation, i.e., random and learning, on GLUE-dev. For random algorithm, 200 categorical distributions are randomly generated for all teacher candidates. For learning algorithm, we employ different learning rates of 3e-4 to 1e-3 with an interval of 1e-4 for categorical distribution optimization. Moreover, the ineffective teacher number is identical to Section 3.3.2 on various tasks. Subsequently, each implementation delivers 25 categorical distributions in phase-2 optimization. Consequently, 200 categorical distributions are obtained. The comparison between random and leaning algorithms is shown in Figure 3.

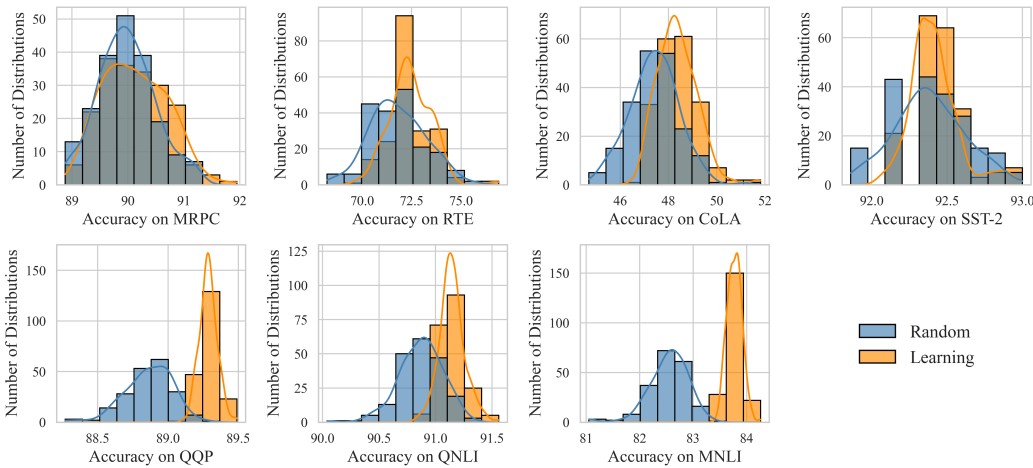

Figure 3: Comparison of AutoSKDBERT with random and learning algorithm for categorical distribution generation on GLUE-dev. Two types of algorithm are evaluated by 200 categorical distributions. MRPC and QQP tasks are evaluated by the average of F1 score and accuracy score, CoLA task is evaluated by Matthews correlation coefficient, and other tasks are evaluated by accuracy score. Best viewed in color.

The categorical distribution generation algorithm aims to achieve more high-performance AutoSKDBERTs. As shown in Figure 3, the proposed learning algorithm contributes to obtaining better categorical distribution than those randomly generated on each downstream task. Particularly, the learning algorithm plays a dominant role on MRPC, CoLA, QQP, QNLI and MNLI tasks. The best accuracy scores of random algorithm are 89.29 on QQP and 83.37 on MNLI, respectively. For the proposed learning algorithm, the worst accuracy scores are 89.11 on QQP and 83.44 on MNLI which rank the top 10 and the best in random algorithm, respectively.

## 5 RELATED WORK

### 5.1 PRE-TRAINED LANGUAGE MODEL

Based on the transformer-style architecture (Vaswani et al., 2017), BERT (Devlin et al., 2019) achieves state-of-the-art performance on different natural language understanding benchmarks, e.g., GLUE (Wang et al., 2019), SQuAD (Rajpurkar et al., 2016; 2018). Subsequently, a great number of variants of BERT are proposed, e.g., XLNet (Yang et al., 2019), ELECTRA (Clark et al., 2020) with new pre-training objectives, RoBERTa (Liu et al., 2019b), T5 (Raffel et al., 2020) with larger pre-training corpus, ConvBERT (Jiang et al., 2020) with various architectures and Synthesizer (Tay et al., 2020) with developed transformer-like block w.r.t. the dot-product self-attention mechanism. Besides, previous pre-trained language models often have several hundred million parameters (e.g. 335 million of BERT$_{LARGE}$ (Devlin et al., 2019), even 175 billion of GPT-3 (Brown et al., 2020)) which contribute to delivering amazing performance on downstream tasks while exponentially increasing the difficulty of deployment on resource-constrained device. ALBERT (Lan et al., 2020) adopts parameter sharing strategy to reduce the parameters, and achieves competitive performance.

### 5.2 KNOWLEDGE DISTILLATION FOR BERT-STYLE LANGUAGE MODEL COMPRESSION

In order to obtain device-friendly BERT-style language model, many KD-based compression approaches have been proposed. DistilBERT (Sanh et al., 2019) compresses a smaller, faster, cheaper and lighter 6-layer BERT-style language model via learning the soft target probabilities of the teacher in the pre-training stage. Sun et al. (2019) propose patient knowledge distillation which transfers knowledge from the last or every $l$ layers, to compress BERT-style language model in the fine-tuning phase. In MobileBERT (Sun et al., 2020), an inverted-bottleneck BERT-style language model is pretrained to transfer knowledge to task-agnostic MobileBERT in a layer-to-layer way. The student in MiniLM (Wang et al., 2020) imitates not only the attention distribution of the teacher, but also the deep self-attention knowledge which reflects the difference between values. In both the pre-training and the fine-tuning phases, TinyBERT (Jiao et al., 2020) learns various knowledge from hidden layer, final layer, embedding and self-attention to achieve high performance. Moreover, GloVe word embedding (Pennington et al., 2014) based data augmentation technique is employed to further improve the performance of TinyBERT. MT-BERT (Wu et al., 2021) employs multiple teachers to achieve better performance than single-teacher KD based approaches on several downstream tasks.

## 6 CONCLUSION

This work proposes AutoSKDBERT, which is a new paradigm of knowledge distillation for BERT model compression. A teacher is stochastically sampled from a predefined multi-level teacher team in each step to distill the student following a categorical distribution. We observe that the categorical distribution plays an important role for obtaining high-performance AutoSKDBERT. Consequently, we propose a two-phase optimization framework to learn the best categorical distribution via SSWO. The first phase distinguishes effective teachers from ineffective teachers. In the second phase, the effective teachers are further optimized. Moreover, before phase-2 optimization beginning, the ineffective teachers are discarded and their weights are assigned to the effective teachers via teacher selection strategy. Extensive experiments on GLUE benchmark show that the proposed AutoSKDBERT achieves state-of-the-art performance compared to popular compression approaches on several downstream tasks.

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

# A INEFFECTIVE TEACHER NUMBER DETERMINATION AND TEACHER TEAM DESIGN

## A.1 INEFFECTIVE TEACHER NUMBER DETERMINATION

A simple way to choose the ineffective teacher number $m$ is elaborately designing the teacher team and setting $m$ to the number of weak teachers whose capacities are weaker than student. Moreover, the student itself can be treated as the above weak teacher.

## A.2 TEACHER TEAM DESIGN

First, we should determine the strongest teacher and student. Next, we select several teacher assistants whose capacities are stronger than student but weaker than the strongest teacher. Finally, we choose also several weak teachers whose capacities are weaker than student. Above all, the pre-defined teacher team consists of several weak teachers, several teacher assistants and the strongest teacher.

# B AUTOSKD FOR IMAGE CLASSIFICATION

To verify the effectiveness of the proposed distillation paradigm on computer vision, we conduct three groups of experiment on CIFAR-100 (Krizhevsky et al., 2009) image classification dataset. Following CRD (Tian et al., 2020), we choose three student models: 1)WRN-16-2, 2) WRN-40-1 and 3) ResNet-8×4. WRN-d-w represents Wide ResNet with depth $d$ and width factor $w$. ResNet-d×4 indicates a 4 times wider network (namely, with 64, 128, and 256 channels for each block) with depth $d$. Moreover, in CRD, the above student models are distilled by WRN-40-2, WRN-40-2 and ResNet-32×4, respectively.

## B.1 DATASET

As a popular dataset for image classification, CIFAR-100 consists of 60000 images (50000 for training and 10000 for test) with 32×32 pixels. Similar to the experiment for BERT compression, the original training set is split fifty-fifty into two subsets, i.e., training subset for student distillation and validation subset for categorical distribution optimization.

## B.2 DETAILS OF TEACHER TEAM

For various student models, we select different teacher teams according to Appendix A.2 as shown in Table 7.

Table 7: Details of teacher team for each student model.

| Student | Teacher Team |
|---|---|
| WRN-16-2 | WRN-16-2, WRN-22-2, WRN-28-2, WRN-34-2, WRN-40-2 |
| WRN-40-1 | WRN-40-1, WRN-16-2, WRN-22-2, WRN-28-2, WRN-34-2, WRN-40-2 |
| ResNet-8×4 | ResNet-8×4, ResNet-14×4, ResNet-20×4, ResNet-26×4, ResNet-32×4 |

Moreover, the performance of each teacher model on CIFAR-100 is shown in Table 8.

Table 8: Performance of each teacher model on CIFAR-100.

| Teacher | WRN-40-1 | WRN-16-2 | WRN-22-2 | WRN-28-2 | WRN-34-2 | WRN-40-2 |
|---|---|---|---|---|---|---|
| Accuracy | 71.19 | 73.18 | 76.19 | 76.21 | 76.14 | 75.61 |

| Teacher | ResNet-8×4 | ResNet-14×4 | ResNet-20×4 | ResNet-26×4 | ResNet-32×4 |
|---|---|---|---|---|---|
| Accuracy | 72.85 | 76.90 | 77.96 | 78.75 | 79.42 |

### B.3 CATEGORICAL DISTRIBUTION OPTIMIZATION

Similar to the experiment for BERT compression, there are two hyper-parameters, i.e., ineffective teacher number and learning rate for categorical distribution optimization. According to Appendix A.1, we fix ineffective teacher number to 1, and choose categorical distribution learning rate from 3e-4 to 1e-3 with an interval of 1e-4 for three groups of experiment. Different from BERT compression, we choose SGD as the optimizer with CosineAnnealing learning rate scheduler, initial learning rate of 0.05 and batch size of 64 for student model training. Moreover, the number of epochs is set to 50, and later 25 epochs deliver 25 categorical distribution candidates.

Table 9: The hyper-parameters for categorical distribution optimization.

| Hyper-parameter | WRN-16-2 | WRN-40-1 | ResNet-8×4 |
|---|---|---|---|
| Ineffective Teacher Number | 1 | 1 | 1 |
| Learning Rate | 1e-3 | 1e-3 | 4e-4 |

### B.4 CATEGORICAL DISTRIBUTION EVALUATION

Following the experimental settings in CRD (Tian et al., 2020), we train the student model with 25 categorical distribution candidates for 240 epochs, and employ SGD as the optimizer with batch size of 64, learning rate of 0.05 which is decayed by a factor of 0.1 when arriving 150-th, 180-th, 210-th epoch and weight decay of 5e-4.

### B.5 LEARNED CATEGORICAL DISTRIBUTION

For various student models, Table 10 shows the learned categorical distributions. Similarly, for each student model, the weakest teacher model, i.e., student itself, is considered as the ineffective teacher model when $m = 1$.

Table 10: Learned categorical distributions for various student models on CIFAR-100.

| Student | WRN-16-2 | | | | | |
|---|---|---|---|---|---|---|
| Teacher | WRN-16-2 | WRN-22-2 | WRN-28-2 | WRN-34-2 | WRN-40-2 | |
| Weight | 0 | 0.2518 | 0.2513 | 0.2458 | 0.2511 | |
| Student | WRN-40-1 | | | | | |
| Teacher | WRN-40-1 | WRN-16-2 | WRN-22-2 | WRN-28-2 | WRN-34-2 | WRN-40-2 |
| Weight | 0 | 0.1864 | 0.2068 | 0.2011 | 0.2034 | 0.2024 |
| Student | ResNet-8×4 | | | | | |
| Teacher | ResNet-8×4 | ResNet-14×4 | ResNet-20×4 | ResNet-26×4 | ResNet-32×4 | |
| Weight | 0 | 0.2299 | 0.2473 | 0.2565 | 0.2663 | |

### B.6 RESULTS AND ANALYSIS

Following Tian et al. (2020), we show the test accuracy of the last epoch in Table 11 for a fair comparison. The proposed distillation paradigm achieves the best performance for 2 out of 3 student models. Particularly, compared to previous state-of-the-art CRD, the improvements are 0.56% and 0.42% for WRN-16-2 and WRN-40-1 distillation, respectively.

## C DETAILS OF STUDENT AND TEACHER TEAM FOR AUTOSKDBERT

### C.1 ARCHITECTURE INFORMATION

The architecture information of student and teachers is shown in Table 12.

### C.2 HYPER-PARAMETERS FOR FINE-TUNING AND DISTILLATION

We utilize the hyper-parameters shown in Table 13 for fine-tuning and distillation.

Table 11: Test accuracy (%) of the proposed AutoSKD and other popular distillation approaches on CIFAR-100. All experimental results are cited from Tian et al. (2020). Average of the last epoch over 5 runs.

| Student | WRN-16-2 | WRN-40-1 | ResNet-8×4 |
|---|---|---|---|
| Teacher | WRN-40-2 | WRN-40-2 | ResNet-32×4 |
| Student Accuracy | 73.26 | 71.98 | 72.50 |
| Teacher Accuracy | 75.61 | 75.61 | 79.42 |
| KD (Hinton et al., 2015) | 74.92 | 73.54 | 73.33 |
| FitNet (Romero et al., 2015) | 73.58 | 72.24 | 73.50 |
| AT (Zagoruyko & Komodakis, 2017) | 74.08 | 72.77 | 73.44 |
| SP (Tung & Mori, 2019) | 73.83 | 72.43 | 72.94 |
| CC (Peng et al., 2019) | 73.56 | 72.21 | 72.97 |
| VID (Ahn et al., 2019) | 74.11 | 73.30 | 73.09 |
| RKD (Park et al., 2019) | 73.35 | 72.22 | 71.90 |
| PKT (Passalis & Tefas, 2018) | 74.54 | 73.45 | 73.64 |
| AB (Heo et al., 2019) | 72.50 | 72.38 | 73.17 |
| FT (Kim et al., 2018) | 73.25 | 71.59 | 72.86 |
| FSP (Yim et al., 2017) | 72.91 | - | 72.62 |
| NST (Huang & Wang, 2017) | 73.68 | 72.24 | 73.30 |
| CRD (Tian et al., 2020) | 75.48 | 74.14 | **75.51** |
| AutoSKD (Ours) | **76.04** | **74.72** | 75.39 |

Table 12: The architecture of each student and teacher.

| Model | Name | Layer | Hidden Size | Head | #Params (M) |
|---|---|---|---|---|---|
| Student | AutoSKDBERT | 6 | 768 | 12 | 66.0 |
| | $T_{01}$ | 8 | 128 | 2 | 5.6 |
| | $T_{02}$ | 10 | 128 | 2 | 6.0 |
| | $T_{03}$ | 12 | 128 | 2 | 6.4 |
| | $T_{04}$ | 8 | 256 | 4 | 14.3 |
| | $T_{05}$ | 10 | 256 | 4 | 15.9 |
| | $T_{06}$ | 12 | 256 | 4 | 17.5 |
| Teacher | $T_{07}$ | 8 | 512 | 8 | 41.4 |
| | $T_{08}$ | 10 | 512 | 8 | 47.7 |
| | $T_{09}$ | 12 | 512 | 8 | 54.0 |
| | $T_{10}$ | 8 | 768 | 12 | 81.1 |
| | $T_{11}$ | 10 | 768 | 12 | 95.3 |
| | $T_{12}$ | 12 | 768 | 12 | 110 |
| | $T_{13}$ | 24 | 1024 | 16 | 335 |
| | $T_{14}$† | 24 | 1024 | 16 | 335 |

† Pre-training with whole word masking.

Table 13: Hyper-parameters for fine-tuning of student and teacher team.

| Hyper-parameter | Value |
|---|---|
| Adam $\epsilon$ | 1e-6 |
| Adam $\beta_1$ | 0.9 |
| Adam $\beta_2$ | 0.999 |
| Learning rate decay | linear |
| Warmup fraction | 0.1 |
| Attention dropout | 0.1 |
| Dropout | 0.1 |
| Weight decay | 1e-4 |
| Batch size | 32 for fine-tuning, {16, 32} for distillation |
| Learning rate | For $T_{13}$ and $T_{14}$, {6e-6, 7e-6, 8e-6, 9e-6} on MRPC and RTE tasks, {2e-5, 3e-5, 4e-5, 5e-5} on other tasks. For student and other teachers, {2e-5, 3e-5, 4e-5, 5e-5} and {1e-5, 2e-5, 3e-5} on all tasks, respectively. |
| Fine-tuning epochs | 15 on MRPC, RTE and CoLA tasks, 5 on other tasks |

## C.3   FINE-TUNING PERFORMANCE

On the one hand, we directly treat the pre-trained model of TinyBERT$_6$[2] as the student of AutoSKD-BERT. On the other hand, we choose 14 BERT-style language models with various capabilities as the candidates for teacher team. Moreover, each pre-trained teacher can be downloaded from official implementation of BERT[3]. Furthermore, the results of the student and the teacher on GLUE-dev are shown in Table 14.

Table 14: The fine-tuning performances of student and teachers on GLUE-dev.

| Model | MRPC $\frac{F1+acc}{2}$ | RTE acc | CoLA Mcc | SST-2 acc | QQP $\frac{F1+acc}{2}$ | QNLI acc | MNLI m | Avg |
|---|---|---|---|---|---|---|---|---|
| Student | 89.44 | 71.84 | 45.74 | 91.63 | 86.44 | 90.70 | 82.55 | 85.87 |
| T$_{01}$ | 81.83 | 66.06 | 25.92 | 86.24 | 83.95 | 83.80 | 72.95 | 80.00 |
| T$_{02}$ | 84.75 | 66.06 | 25.57 | 85.67 | 84.18 | 84.00 | 73.75 | 80.49 |
| T$_{03}$ | 84.59 | 65.70 | 27.87 | 86.47 | 85.02 | 84.40 | 75.16 | 81.01 |
| T$_{04}$ | 85.18 | 64.62 | 40.35 | 89.33 | 86.36 | 86.80 | 78.16 | 82.46 |
| T$_{05}$ | 87.84 | 66.06 | 38.76 | 89.33 | 87.25 | 87.26 | 78.75 | 83.42 |
| T$_{06}$ | 85.96 | 66.06 | 41.36 | 89.68 | 87.21 | 87.42 | 79.54 | 83.27 |
| T$_{07}$ | 87.91 | 70.04 | 48.14 | 91.28 | 88.69 | 89.27 | 80.84 | 85.21 |
| T$_{08}$ | 88.17 | 65.70 | 50.98 | 91.28 | 88.62 | 89.25 | 81.41 | 84.74 |
| T$_{09}$ | 88.85 | 66.43 | 53.58 | 92.09 | 89.01 | 90.33 | 81.90 | 85.34 |
| T$_{10}$ | 89.36 | 68.95 | 56.30 | 93.00 | 89.27 | 90.79 | 83.05 | 86.21 |
| T$_{11}$ | 90.10 | 71.12 | 60.32 | 92.78 | 89.71 | 91.20 | 84.00 | 86.93 |
| T$_{12}$ | 89.98 | 68.59 | 60.26 | 92.66 | 89.66 | 91.85 | 84.40 | 86.76 |
| T$_{13}$ | **90.60** | 62.74 | 62.74 | 94.50 | 90.26 | 92.70 | 86.88 | 86.83 |
| T$_{14}$ | 90.15 | **79.06** | **65.88** | **94.72** | **90.40** | **93.89** | **87.06** | **89.50** |

## C.4   PERFORMANCE OF STUDENT WITH VARIOUS DISTILLATION PARADIGMS

Table 15 summarizes the performance of student using different distillation paradigms with the teacher models described in Appendix C.1. Moreover, experimental settings can be found in Table 13. On the one hand, the student performance using single-teacher distillation with respect to each teacher model is given. On the other hand, two popular multi-teacher KD paradigms, i.e., AvgKD (Hinton et al., 2015) and TAKD (Mirzadeh et al., 2020), are employed to distill the student with two groups of teacher team, i.e., T$_{01}$ to T$_{14}$ and T$_{10}$ to T$_{14}$.

According to Table 15, we can draw several conclusions:

1. For single-teacher KD paradigm, the strongest teacher may not be the best teacher for student distillation. Capacity gap (Mirzadeh et al., 2020) between the strong-capacity teacher and weak-capacity student plays an important role for this phenomenon.

2. For multi-teacher AvgKD, increasing the number of teachers can not always contribute to improving the distillation performance. In AvgKD, the diversity losing issue leads to unsatisfactory performance due to using the ensemble of teacher outputs.

3. For multi-teacher TAKD, weak-capacity teachers dramatically reduce the distillation performance of student. In TAKD, the weakest teacher assistant (e.g., T$_{01}$ for the teacher team T$_{01}$-T$_{14}$, T$_{10}$ for the teacher team T$_{10}$-T$_{14}$) transfers mixture of knowledge which learned from previous stronger teacher assistants (e.g., T$_{02}$ to T$_{14}$ for the teacher team T$_{01}$-T$_{14}$, T$_{11}$ to T$_{14}$ for the teacher team T$_{10}$-T$_{14}$) into the student. As a result, the performance of TAKD is very sensitive to the capacity of the weakest teacher assistant.

In order to verify the effectiveness of weak-capacity teacher for performance improvement, we choose several weak-capacity BERT-style models as teachers, e.g., T$_{01}$ to T$_{09}$. Besides, we choose also two strong-capacity teachers, i.e., T$_{13}$ and T$_{14}$ in Table 12, to verify the effectiveness of the proposed distillation paradigm for capacity gap alleviation.

---

[2]https://huggingface.co/huawei-noah/TinyBERT_General_6L_768D
[3]https://github.com/google-research/bert

Table 15: Distillation performance of student with various distillation paradigms on GLUE-dev.

| KD Paradigm | Teacher | MRPC $\frac{F1+acc}{2}$ | RTE acc | CoLA Mcc | SST-2 acc | QQP $\frac{F1+acc}{2}$ | QNLI acc | MNLI m |
|---|---|---|---|---|---|---|---|---|
| Single-teacher | $T_{01}$ | 84.6 | 67.9 | 32.5 | 88.8 | 84.6 | 86.3 | 74.9 |
| | $T_{02}$ | 87.9 | 67.2 | 35.3 | 89.7 | 85.1 | 86.2 | 75.3 |
| | $T_{03}$ | 87.1 | 70.8 | 36.7 | 90.5 | 86.1 | 86.5 | 76.2 |
| | $T_{04}$ | 89.7 | 69.0 | 40.2 | 92.4 | 87.1 | 89.6 | 79.7 |
| | $T_{05}$ | 89.7 | 71.1 | 46.9 | 91.4 | 87.8 | 89.8 | 79.8 |
| | $T_{06}$ | 88.4 | 69.3 | 45.8 | 92.4 | 87.6 | 90.2 | 80.4 |
| | $T_{07}$ | 89.5 | 73.7 | 48.3 | 92.8 | 88.6 | 91.0 | 81.9 |
| | $T_{08}$ | 89.6 | 71.5 | 46.7 | 92.3 | 88.8 | 90.9 | 82.3 |
| | $T_{09}$ | 89.9 | 72.2 | 46.2 | 92.2 | 88.9 | 91.5 | 82.7 |
| | $T_{10}$ | 89.6 | 73.3 | 49.3 | 92.0 | 88.9 | 91.1 | 82.9 |
| | $T_{11}$ | 89.7 | 71.8 | 48.5 | 92.3 | 89.0 | 91.3 | 83.2 |
| | $T_{12}$ | 89.1 | 71.5 | 46.9 | **93.1** | 88.9 | 91.4 | 82.8 |
| | $T_{13}$ | 90.0 | 72.9 | 47.7 | 92.1 | 88.9 | 91.2 | 83.4 |
| | $T_{14}$ | 89.5 | 72.6 | 48.3 | 92.4 | 89.0 | 91.3 | 83.5 |
| AvgKD (Hinton et al., 2015) | $T_{01}$-$T_{14}$ | 90.2 | 71.8 | 47.2 | 92.2 | 89.1 | 91.1 | 83.5 |
| | $T_{10}$-$T_{14}$ | 89.9 | 72.9 | 48.4 | 92.2 | 89.0 | 91.2 | 83.4 |
| TAKD (Mirzadeh et al., 2020) | $T_{01}$-$T_{14}$ | 83.7 | 67.9 | 29.4 | 88.0 | 83.2 | 84.6 | 73.6 |
| | $T_{10}$-$T_{14}$ | 89.3 | 71.8 | 47.8 | 92.7 | 88.7 | 91.4 | 83.4 |
| AutoSKDBERT (Ours) | $T_{01}$-$T_{14}$ | **92.0** | **76.9** | **51.8** | 93.0 | **89.5** | **91.6** | **84.3** |

# D ORTHOGONAL EXPERIMENT OF AUTOSKDBERT WITH TRANSFORMER LAYER DISTILLATION AND DATA AUGMENTATION

This paper proposes a general distillation paradigm for BERT compression. Consequently, most of other distillation approaches can combine with the proposed distillation paradigm. For instance, transformer layer distillation used in TinyBERT (Jiao et al., 2020) and MoEBERT (Zuo et al., 2022) can be replaced with the stochastic KD paradigm proposed in this paper. Moreover, each teacher in the teacher team should has same hidden size with the student when distilling the transformer layer. Consequently, we can not distill the student with the teacher team used in this paper. In this section, we implement a list of orthogonal experiments to examine the effectiveness of the combination of AutoSKDBERT and TinyBERT, and show the experimental result in Table 16.

Table 16: Results of AutoSKDBERT with Data Augmentation (DA) transformer layer Distillation (TD) on GLUE-dev.

| Model | DA | TD | MRPC $\frac{F1+acc}{2}$ | RTE acc | CoLA Mcc | SST-2 acc | QNLI acc | Avg |
|---|---|---|---|---|---|---|---|---|
| AutoSKDBERT | ✗ | ✗ | **92.0** | **76.9** | 51.8 | **93.0** | 91.6 | **81.1** |
| | ✓ | ✗ | 89.7 | 73.3 | **56.8** | 92.9 | **92.0** | 80.9 |
| | ✗ | ✓ | 87.8 | 70.0 | 41.1 | 92.2 | 91.1 | 76.4 |
| | ✓ | ✓ | 91.1 | 69.3 | 55.3 | 92.7 | **92.0** | 80.1 |

Due to the difference of hidden size between the strongest teacher $T_{14}$ and the student, similar to TinyBERT, we employ also BERT$_{BASE}$, i.e., $T_{12}$, as the teacher for transformer layer distillation. TinyBERT employs random search to choose the best batch size and learning rate from {16, 32} and {1e-5, 2e-5, 3e-5}, respectively. Differently, AutoSKDBERT uses also the categorical distributions on vanilla datasets with the batch size and the learning rate shown in Table 2 for each downstream task. Moreover, the epoch number for each downstream task can be found in Table 13.

As shown in Table 16, the combination of AutoSKDBERT and DA shows better performance compared to vanilla AutoSKDBERT on CoLA and QNLI tasks. Furthermore, the combination of AutoSKDBERT, TD and DA achieves better performance compared to vanilla AutoSKDBERT on QNLI tasks. The main reason in our consideration is that the categorical distributions are learned on the vanilla dataset instead of the augmentation data. In the future, we will directly learn the categorical distribution on the augmentation data.

However, the combination of AutoSKDBERT and TD is prone to obtaining worse performance on each downstream task. We consider that the main cause of the above phenomenon is the knowledge transfer gap between transformer layer distillation and prediction layer distillation, i.e., only using $T_{12}$ for transformer layer distillation, $T_{01}$ to $T_{14}$ for prediction layer distillation. In the future, we will select appropriate teacher team to distill the transformer layer of BERT.

# E    IMPACT OF HYPER-PARAMETERS FOR CATEGORICAL DISTRIBUTION OPTIMIZATION

As above mentioned, there are two important hyper-parameters, i.e., ineffective teacher number $m$ and learning rate, for categorical distribution optimization. In this section, we discuss the impact of the above two hyper-parameters for the best performance of the learned categorical distributions. On the tasks of MRPC, RTE and CoLA, we implement AutoSKDBERT with $m$ from 1 to 10 and learning rate from 3e-4 to 1e-3 with an interval of 1e-4, and show the results in Table 17.

Table 17: Results of AutoSKDBERT with various hyper-parameters for categorical distribution optimization.

| Task | Metric | $m$ | Learning Rate | | | | | | | | Mean±Std |
|---|---|---|---|---|---|---|---|---|---|---|---|
| | | | 3e-4 | 4e-4 | 5e-4 | 6e-4 | 7e-4 | 8e-4 | 9e-4 | 1e-3 | |
| MRPC | $\frac{F1+acc}{2}$ | 1 | 90.3 | 90.6 | 91.4 | 90.8 | 91.1 | 91.0 | 92.0 | 91.6 | 91.1±0.6 |
| | | 2 | 91.7 | 91.6 | 91.6 | 91.8 | 91.6 | 91.6 | 91.5 | 91.8 | 91.7±0.1 |
| | | 3 | 90.5 | 90.9 | 91.1 | 91.1 | 90.9 | 91.2 | 90.6 | 90.8 | 90.9±0.2 |
| | | 4 | 90.3 | 90.1 | 90.6 | 91.0 | 91.3 | 90.4 | 91.0 | 90.4 | 90.6±0.4 |
| | | 5 | 90.7 | 91.1 | 90.7 | 90.7 | 91.8 | 90.3 | 90.8 | 90.4 | 90.8±0.4 |
| | | 6 | 90.6 | 90.6 | 90.4 | 90.2 | 90.8 | 90.5 | 89.9 | 90.9 | 90.5±0.3 |
| | | 7 | 90.7 | 90.2 | 91.1 | 90.6 | 90.1 | 90.4 | 89.8 | 90.5 | 90.4±0.4 |
| | | 8 | 90.8 | 90.7 | 90.2 | 90.5 | 90.5 | 90.6 | 90.5 | 90.3 | 90.5±0.2 |
| | | 9 | 90.4 | 90.1 | 90.9 | 89.9 | 90.1 | 90.8 | 90.6 | 90.3 | 90.4±0.3 |
| | | 10 | 90.4 | 89.8 | 89.8 | 90.4 | 89.9 | 91.1 | 90.7 | 90.4 | 90.3±0.4 |
| **Mean±Std** | | | 90.6±0.4 | 90.6±0.5 | 90.8±0.5 | 90.7±0.5 | 90.8±0.6 | 90.8±0.4 | 90.7±0.6 | 90.7±0.5 | |
| RTE | acc | 1 | 73.3 | 72.6 | 73.7 | 72.6 | 72.6 | 72.6 | 72.2 | 73.3 | 72.9±0.5 |
| | | 2 | 72.9 | 72.9 | 73.7 | 72.9 | 73.7 | 73.3 | 74.7 | 75.5 | 73.7±0.9 |
| | | 3 | 72.9 | 76.9 | 73.7 | 73.7 | 72.9 | 74.4 | 74.4 | 74.0 | 74.1±1.2 |
| | | 4 | 72.6 | 72.9 | 74.0 | 73.3 | 76.2 | 73.7 | 75.1 | 73.3 | 73.9±1.1 |
| | | 5 | 74.4 | 74.7 | 74.0 | 73.3 | 74.7 | 74.0 | 74.7 | 75.5 | 74.4±0.6 |
| | | 6 | 75.5 | 76.2 | 74.0 | 74.7 | 75.5 | 75.1 | 76.2 | 76.2 | 75.4±0.7 |
| | | 7 | 74.1 | 75.1 | 72.2 | 74.4 | 74.4 | 74.0 | 74.0 | 74.4 | 74.1±0.8 |
| | | 8 | 75.8 | 73.7 | 74.4 | 74.4 | 74.0 | 74.7 | 74.4 | 74.4 | 74.5±0.6 |
| | | 9 | 74.0 | 73.3 | 73.7 | 73.7 | 74.4 | 74.4 | 74.4 | 76.9 | 74.4±1.0 |
| | | 10 | 72.9 | 74.7 | 72.6 | 75.1 | 73.7 | 73.7 | 74.7 | 74.0 | 73.9±0.8 |
| **Mean±Std** | | | 73.8±1.1 | 74.3±1.4 | 73.6±0.6 | 73.8±0.8 | 74.2±1.0 | 74.0±0.7 | 74.5±0.9 | 74.8±1.2 | |
| CoLA | Mcc | 1 | 50.2 | 50.0 | 50.0 | 50.0 | 49.7 | 49.8 | 50.6 | 49.6 | 50.0±0.3 |
| | | 2 | 50.0 | 50.9 | 49.8 | 50.3 | 50.2 | 49.8 | 49.2 | 50.2 | 50.0±0.5 |
| | | 3 | 48.3 | 48.6 | 49.4 | 49.8 | 50.6 | 50.2 | 50.1 | 49.9 | 49.6±0.7 |
| | | 4 | 49.9 | 49.3 | 49.8 | 50.8 | 51.8 | 50.5 | 49.4 | 51.5 | 50.4±0.9 |
| | | 5 | 49.2 | 49.0 | 49.3 | 50.0 | 49.7 | 49.3 | 49.4 | 49.3 | 49.4±0.3 |
| | | 6 | 48.7 | 49.3 | 50.0 | 49.3 | 49.5 | 49.7 | 49.8 | 49.4 | 49.5±0.4 |
| | | 7 | 49.3 | 48.7 | 50.0 | 49.3 | 49.7 | 49.7 | 49.2 | 49.0 | 49.4±0.4 |
| | | 8 | 49.6 | 49.3 | 49.1 | 49.4 | 49.9 | 50.5 | 50.6 | 50.2 | 49.8±0.5 |
| | | 9 | 49.8 | 49.7 | 50.1 | 49.0 | 49.2 | 48.7 | 50.2 | 49.6 | 49.5±0.5 |
| | | 10 | 49.2 | 49.2 | 49.6 | 48.5 | 50.6 | 49.7 | 49.4 | 48.6 | 49.4±0.6 |
| **Mean±Std** | | | 49.4±0.6 | 49.4±0.6 | 49.7±0.3 | 49.6±0.6 | 50.1±0.7 | 49.8±0.5 | 49.8±0.5 | 49.7±0.8 | |

We can draw a conclusion from Table 17 that the proposed AutoSKDBERT is sensitive to the ineffective teacher number rather than the learning rate. For each task, the ineffective teacher number $m$ plays a more important role compared to the learning rate for AutoSKDBERT. For instance, the mean value of $m = 2$ is 91.7 which is 1.4 higher than the mean value of $m = 10$. However, the largest difference of mean value with respect to learning rate is only 0.2. There is a similar phenomenon on the tasks of RTE and CoLA. Therefore, the above conclusion can be drawn.

## F  COST COMPARISON OF TINYBERT AND AUTOSKDBERT

In this section, we show the cost of AutoSKDBERT in terms of categorical distribution optimization and evaluation, and compare our approach to TinyBERT with respect to algorithm cost. Experimental results are shown in Table 18 where on five downstream tasks, the cost of AutoSKDBERT is 38.72 hours which is 8.4× less than TinyBERT. Moreover, we obtain the cost on NVIDIA A100 GPU with AMD EPYC 7642 48-Core Processor.

Table 18: The cost (hours) comparison of TinyBERT and AutoSKDBERT on five downstream tasks. These results about TinyBERT are obtained by following the experimental settings described in Jiao et al. (2020) with the code publicly released by the authors at https://github.com/huawei-noah/Pretrained-Language-Model/tree/master/TinyBERT.

|  | TinyBERT | | | | | AutoSKDBERT | | | | |
| --- | --- | --- | --- | --- | --- | --- | --- | --- | --- | --- |
|  | MRPC | RTE | CoLA | SST-2 | QNLI | MRPC | RTE | CoLA | SST-2 | QNLI |
| Transformer Layer Distillation | 3.42 | 2.80 | 12.72 | 12.90 | 61.94 | 0 | 0 | 0 | 0 | 0 |
| Categorical Distribution Optimization | 0 | 0 | 0 | 0 | 0 | 0.26 | 0.17 | 1.06 | 1.28 | 2.45 |
| Prediction Layer Distillation† | 5.04 | 3.24 | 4.35 | 31.71 | 187.62 | 1.75 | 1.25 | 5.75 | 6.50 | 18.25 |
| Total Cost | 8.46 | 6.04 | 17.07 | 44.61 | 249.56 | 2.01 | 1.42 | 6.81 | 7.78 | 20.70 |

† For TinyBERT, the cost is obtained by 6 groups of experiment with various hyper-parameters (i.e., batch sizes of {16, 32} and learning rates of {1e-5, 2e-5, 3e-5}) on augmentation data. For AutoSKDBERT, the cost is obtained by 25 groups of experiment on vanilla data with different categorical distributions learned in the process of categorical distribution optimization.

The distillation process of TinyBERT can be divided into two phases: 1) transformer layer distillation on augmentation data and 2) prediction layer distillation on augmentation data. The transformer layer distillation of TinyBERT is time-consuming, e.g., it spends about 62 hours on QNLI. Besides, the prediction layer distillation of TinyBERT is also time-consuming due to using large-scale augmentation data.

Differently, AutoSKDBERT consists of categorical distribution optimization and evaluation (i.e., prediction layer distillation). On the one hand, categorical distribution optimization is efficient, e.g., 2.45 hours on the task of QNLI, due to the gradient-based SSWO. On the other hand, categorical distribution evaluation is also efficient even choosing the best categorical distribution from 25 candidates.

## G  TINYBERT WITH STRONGER TEACHER MODEL

AutoSKDBERT employs two stronger teacher models, i.e., $T_{13}$ and $T_{14}$, compared to most of the comparative methods shown in Table 4. To verify the impact of strong teacher on the distillation performance of other paradigms, we employ $T_{12}$ and $T_{14}$ as the teachers to distill TinyBERT on five downstream tasks for a fair comparison. Following TinyBERT (Jiao et al., 2020), we implement the experiments with batch sizes of {16, 32} and learning rates of {1e-5, 2e-5, 3e-5}, and choose the best result to show in Table 19.

Table 19: Results of TinyBERT with the strongest teacher $T_{14}$ on GLUE-dev. These results are obtained by TinyBERT with the fine-tuned teacher model of AutoSKDBERT using the code publicly released by the authors at https://github.com/huawei-noah/Pretrained-Language-Model/tree/master/TinyBERT.

| Student | Teacher | MRPC $\frac{F1+acc}{2}$ | RTE acc | CoLA Mcc | SST-2 acc | QNLI acc | Avg |
| --- | --- | --- | --- | --- | --- | --- | --- |
| TinyBERT (w/o aug) | $T_{12}$ | 87.0 | 67.9 | 42.2 | 92.0 | 91.2 | 76.1 |
| | $T_{14}$ | 86.7 | 70.0 | 40.9 | 92.0 | 90.9 | 76.1 |

We can observe that the strong teacher $T_{14}$ contributes to only improving the performance on RTE. For the above phenomenon, the main reason is that a capacity gap (Mirzadeh et al., 2020) exists between $T_{14}$ and student which is prone to obtaining unsatisfactory performance. As a result, a conclusion can be drawn that the stronger teacher $T_{14}$ can not always contribute to improving the performance of other distillation paradigms.

## H MT-BERT FOR BERT COMPRESSION

For BERT-style language model compression, we verify the performance of MT-BERT (Wu et al., 2021) whose objection function can be expressed as:

$$\mathcal{L}_{\text{MTBERT}} = \sum_{i=1}^{N} \frac{\text{CE}(y_i/T, y_s/T)}{1 + \text{CE}(y, y_i)}, \tag{9}$$

where, $N$ indicates the number of teachers, $\text{CE}(\cdot, \cdot)$ is the cross-entropy loss, $T$ denotes the temperature, $y$ represents the ground-truth label, $y_i$ and $y_s$ refer to the outputs of $i$-th teacher and the student, respectively.

We employ $\text{T}_{10}$ to $\text{T}_{14}$ as the teacher team to distill the student via Eq. 9. Particularly, we only use the weighted multi-teacher distillation loss without the multi-teacher hidden loss and the task-specific loss as in MT-BERT (Wu et al., 2021).

The hyper-parameters are given as follows:

- **Learning Rate**: {1e-5, 2e-5, 3e-5} for all tasks.
- **Batch Size**: {16, 32, 64}.
- **Epoch**: 10 for MRPC, RTE and CoLA tasks, 3 for other tasks.

Other settings follow AutoSKDBERT.

## I DETAILS OF GLUE BENCHMARK

GLUE consists of 9 NLP tasks: Microsoft Research Paraphrase Corpus (MRPC) (Dolan & Brockett, 2005), Recognizing Textual Entailment (RTE) (Bentivogli et al., 2009), Corpus of Linguistic Acceptability (CoLA) (Warstadt et al., 2019), Semantic Textual Similarity Benchmark (STS-B) (Cer et al., 2017), Stanford Sentiment Treebank (SST-2) (Socher et al., 2013), Quora Question Pairs (QQP) (Chen et al., 2018), Question NLI (QNLI) (Rajpurkar et al., 2016), Multi-Genre NLI (MNLI) (Williams et al., 2017), and Winograd NLI (WNLI) (Levesque et al., 2012).

**MRPC** belongs to a sentence similarity task where system aims to identify the paraphrase/semantic equivalence relationship between two sentences.

**RTE** belongs to a natural language inference task where system aims to recognize the entailment relationship of given two text fragments.

**CoLA** belongs to a single-sentence task where system aims to predict the grammatical correctness of an English sentence.

**STS-B** belongs to a sentence similarity task where system aims to evaluate the similarity of two pieces of texts by a score from 1 to 5.

**SST-2** belongs to a single-sentence task where system aims to predict the sentiment of movie reviews.

**QQP** belongs to a sentence similarity task where system aims to identify the semantical equivalence of two questions from the website Quora.

**QNLI** belongs to a natural language inference task where system aims to recognize that for a given pair *<question, context>*, the answer to the *question* whether contains in the *context*.

**MNLI** belongs to a natural language inference task where system aims to predict the possible relationships (i.e., entailment, contradiction and neutral) of *hypothesis* w.r.t. *premise* for a given pair *<premise, hypothesis>*.

**WNLI**   belongs to a natural language inference task where system aims to determine the referent of a sentence's pronoun from a list of choices.

