# OpenReview forum: "AutoSKDBERT: Learn to Stochastically Distill BERT"
_ICLR.cc/2023/Conference — Submitted to ICLR 2023_

### Official Review · Reviewer_FHc1 · 2022-10-20

**Confidence:** 3
**Correctness:** 4
**Technical Novelty And Significance:** 3
**Empirical Novelty And Significance:** 3
**Recommendation:** 5

**Clarity, Quality, Novelty And Reproducibility:**

Clarity:

Overall the clarity is good.

Some questions:
In the second paragraph in Introduction, "the ensemble of multiple teachers are not always more effective than the single teacher for student distillation........ On the other hand, between the large-capability teacher and small-capability student, there is a capability gap which can be prone to unsatisfactory distillation performance" I wonder why the capability gap of student and teacher can be used to explain why the *ensemble* of multiple teacher? Is it because the capacity is low in student, so adding more capacity (by ensemble) does not help?

In table 1, I am a bit confused why different data set uses so many different metric. It would be helpful to report 2-3 metrics for all dataset, to avoid potential cherry picking.


Quality, Novelty


Here are some questions:
It would be also interesting to see the training time for distilling of the proposed method compared to other existed method.

Giveh there are multiple metrics, it is hard for readers to understand the magnitude of the improvement. It would be helpful to add information like standard error/confidence interval for each metric.

For the comparison benchmark, which of them are using multiple teachers? It is important to have a fair comparison given the improvement can come both from the proposed distilling as well as more teachers. I would recommend add some simple heuristic based distillation method with multi-teacher as benchmark.


Reproducibility

Overall the experiment description is clear, and data used in experiments are open-sourced public data. However, it may still hard to reproduce without the experiment script.

For "The hyper-parameters for student distillation.", it would be helpful to determine why different dataset has different parameter. Are those hyper-parameters recommended by other research?

*More important*, "The hyper-parameters for categorical distribution optimization." how such parameter are selected? Is the algorithm sensitive to such parameters?

Also it may be helpful to add how multi-teacher is used? Is it just use all teachers in all examples? As there are so many distill methods, it would be helpful to add details for reproducibility.

**Strength And Weaknesses:**

Strength
Authors proposed a novel learning algorithm to learn from multiple teachers.

Weaknesses
This paper only focused on bert. My understanding is the proposed method can be adapted to other network architecture. The impact would be larger if such method is independent of the network architecture of teacher/student.

**Summary Of The Paper:**

In this paper, authors proposed AutoSKDBERT, a new knowledge distillation paradigm for BERT compression, that stochastically samples a teacher from a predefined teacher team following a categorical distribution in each step, to transfer knowledge into student.

With extensive experiments on GLUE benchmark show that the proposed AutoSKDBERT achieves state-of-the-art score compared to previous compression approaches on several downstream tasks.

**Summary Of The Review:**

I choose the final decision 'marginally below' given my questions above as well as the proposed method for multi-teacher is only studied on BERT. If authors are able to clarify and extend the work, I would be happy to reconsider the decision.

---

> ### Author Response · Authors · 2022-11-19
> **Part 1 of Response to Reviewer FHc1**
>
> We thank you for your comments and feedback. In addition to the general updates, we address your concerns here.
>
> **Q1: This paper only focused on bert.**
>
> As described in General Response, we have added the experiment for image classification on CIFAR-100 (see Appendix B in the revised manuscript), to verify the generality of the proposed distillation paradigm. Experimental results show that the proposed approach can also achieve novel performance for CNN-based image classification.
>
> **Q2: In the second paragraph in Introduction, "the ensemble of multiple teachers are not always more effective than the single teacher for student distillation........ On the other hand, between the large-capability teacher and small-capability student, there is a capability gap which can be prone to unsatisfactory distillation performance" I wonder why the capability gap of student and teacher can be used to explain why the ensemble of multiple teacher? Is it because the capacity is low in student, so adding more capacity (by ensemble) does not help?**
>
> A2: We are very sorry to confuse you due to our writing mistake. We aims to express that there is a capacity gap between large-capability teacher ensemble (instead of the single teacher) and small-capability student. In the second paragraph in Introduction of the revised manuscript, we have modified the expression as "On the one hand, the ensemble prediction of multi-teacher KD loses the diversity of each teacher. On the other hand, between the large-capacity teacher ensemble and small-capacity student, there is a capacity gap which can be prone to unsatisfactory distillation performance".
>
> **Q3: In table 1, I am a bit confused why different data set uses so many different metric. It would be helpful to report 2-3 metrics for all dataset, to avoid potential cherry picking.**
>
> A3: The used 7 datasets belong to different tasks (more details can be found in Appendix I in the revised manuscript). Consequently, following previous works [1, 2], we employ identical metrics for each task as shown in Table 1.
>
> **Q4: It would be also interesting to see the training time for distilling of the proposed method compared to other existed method.**
>
> A4: We have added the results of the cost of AutoSKDBERT, and compared to previous SOTA TinyBERT [2] (see Appendix F in the revised manuscript). Experimental results show that the cost of our approach is 8.4$\times$ less than TinyBERT.
>
> **Q5: Given there are multiple metrics, it is hard for readers to understand the magnitude of the improvement. It would be helpful to add information like standard error/confidence interval for each metric.**
>
> A5: Indeed, the standard error and confidence are helpful to understand the magnitude of the improvement. In this paper, following previous works, we do not show the above information.
>
> **Q6: For the comparison benchmark, which of them are using multiple teachers? It is important to have a fair comparison given the improvement can come both from the proposed distilling as well as more teachers. I would recommend add some simple heuristic based distillation method with multi-teacher as benchmark.**
>
> A6:
>
> For the comparison benchmarks, only MT-BERT uses multiple teachers.
>
> We have added the distillation results of 1) single-teacher KD, 2) multi-teacher AvgKD [3] and 3) multi-teacher TAKD [4] (see Appendix C.4 in the revised manuscript), to make a fair comparison with AutoSKDBERT. Experimental results show that 1) AutoSKDBERT achieves the best performance on all tasks compared to AvgKD and TAKD, 2) a conclusion can be drawn that extreme strong teacher can not always contribute to improving the distillation performance.
>
>  **Q7: Overall the experiment description is clear, and data used in experiments are open-sourced public data. However, it may still hard to reproduce without the experiment script.**
>
> A7: The code will be made publicly available upon publication of the paper.
>
> **Q8: For "The hyper-parameters for student distillation.", it would be helpful to determine why different dataset has different parameter. Are those hyper-parameters recommended by other research?**
>
> A8: Yes, for student distillation, the used hyper-parameters are recommended by previous works, e.g., TinyBERT [2], MT-BERT [5], etc.
>
> **Q9: "The hyper-parameters for categorical distribution optimization." how such parameter are selected? Is the algorithm sensitive to such parameters?**
>
> A9: We select the ineffective teacher number and the learning rate from {1,2,...,10} and {3e-4, 4e-4,...,1e-3}, respectively. We have shown the performances of AutoSKDBERT using various hyper-parameters (see Appendix D in the revised manuscript). Experimental results show that AutoSKDBERT is sensitive to the ineffective teacher number rather than the learning rate.

---

> > ### Author Response · Authors · 2022-11-19
> > **Part 2 of Response to Reviewer FHc1**
> >
> > **Q10: Also it may be helpful to add how multi-teacher is used? Is it just use all teachers in all examples? As there are so many distill methods, it would be helpful to add details for reproducibility.**
> >
> > For AutoSKDBERT, we use $T_{01}$ to $T_{14}$ for all tasks where the sampling weights of ineffective teachers are zero. For other multi-teacher KD paradigms, we have added the used teachers in the revised manuscript.
> >
> > [1] Devlin J, Chang M W, Lee K, et al. Bert: Pre-training of deep bidirectional transformers for language understanding[J]. arXiv preprint arXiv:1810.04805, 2018.
> >
> > [2] Jiao X, Yin Y, Shang L, et al. Tinybert: Distilling bert for natural language understanding[J]. arXiv preprint arXiv:1909.10351, 2019.
> >
> > [3] Hinton G, Vinyals O, Dean J. Distilling the knowledge in a neural network[J]. arXiv preprint arXiv:1503.02531, 2015, 2(7).
> >
> > [4] Mirzadeh S I, Farajtabar M, Li A, et al. Improved knowledge distillation via teacher assistant[C]//Proceedings of the AAAI conference on artificial intelligence. 2020, 34(04): 5191-5198.
> >
> > [5] Wu C, Wu F, Huang Y. One teacher is enough? pre-trained language model distillation from multiple teachers[J]. arXiv preprint arXiv:2106.01023, 2021.
> >
> > Thank you for the feedback, we hope we have adequately addressed your concerns. We will be happy to answer any additional questions. We encourage you to reconsider your score in light of our updates.

---

### Official Review · Reviewer_SQHt · 2022-10-20

**Confidence:** 5
**Correctness:** 3
**Technical Novelty And Significance:** 2
**Empirical Novelty And Significance:** 2
**Recommendation:** 5

**Clarity, Quality, Novelty And Reproducibility:**

The formulation of the problem is clear and the paper is well-written. However, the novelty is quite limited. The paper presents experimental details for reproducibility.

**Strength And Weaknesses:**

**Strength**:

1. The formulation of the problem is clear and the paper is well-written.
2. Most relevant works are discussed. The experiments show that AutoSKDBERT achieves good results.

**Weaknesses**:
1. To show that optimizing the categorical distribution is necessary, I wonder whether a single-teacher KD is enough. Table 1 shows some results. However, I would like to see the complete results from T1 to T14. Besides, does "Best Single Teacher" refers to the teacher that can achieve the best performance for student distillation or the teacher itself that achieves the best performance?
2. The novelty of the proposed method is limited.

**Summary Of The Paper:**

Given a predefined teacher team, the paper aims at finding an optimal categorical distribution of these teacher models for student network distillation. Rather than simple ensemble learning (uniform weight), the paper proposes a two-stage selection and optimization strategy. In the first stage, ineffective teachers are identified and discarded by optimizing the categorical distribution. In the second stage, the categorical distribution is further tuned for effective teacher teams. The paper conducts experiments on the GLUE benchmark and make comparison with the most relevant works such as DistilBERT and TinyBERT.

**Summary Of The Review:**

Although the proposed method achieves decent results, the technical novelty is quite limited. Optimizing the categorical choice while optimizing the network parameters is a widely studied bilevel optimization problem in neural architecture search and other areas. Another important concern is that a more simple baseline should be considered such as single best teacher and Gumbel softmax optimization to systematically study whether it is necessary to optimize the categorical distribution.

---

> ### Author Response · Authors · 2022-11-19
> **Response to Reviewer SQHt**
>
> We thank you for your comments and feedback. In addition to the general updates, we address your concerns here.
>
> **Q1: To show that optimizing the categorical distribution is necessary, I wonder whether a single-teacher KD is enough. Table 1 shows some results. However, I would like to see the complete results from T1 to T14. Besides, does "Best Single Teacher" refers to the teacher that can achieve the best performance for student distillation or the teacher itself that achieves the best performance?**
>
> A1:
>
> **To show that optimizing the categorical distribution is necessary, I wonder whether a single-teacher KD is enough. Table 1 shows some results. However, I would like to see the complete results from T1 to T14.**
>
> We have added the experiments of single-teacher KD using $T_{01}$ to $T_{14}$ as the teacher (see Appendix C.4 in the revised manuscript). As shown in Table 15, AutoSKDBERT achieves the best performance on 6 out of 7 tasks, compared to $T_{01}$ to $T_{14}$ based single-teacher KD, multi-teacher AvgKD [1] and TAKD [2]. Moreover, we have shown the performance comparison of AutoSKDBERT with random and optimized categorical distributions in Section 4.3. Experimental results show that the optimized categorical distributions achieve higher performance than those randomly generated categorical distributions for all tasks. Above all, it is necessary to optimize the categorical distribution.
>
> **Does "Best Single Teacher" refers to the teacher that can achieve the best performance for student distillation or the teacher itself that achieves the best performance?**
>
> Yes, "Best Single Teacher" in Table 1 refers to the teacher that can achieve the best performance for student distillation. Moreover, the student distillation performances with respect to 14 teachers ($T_{01}$ to $T_{14}$) have been added in the revised manuscript (see Appendix C.4).
>
> **Q2: The novelty of the proposed method is limited.**
>
> A2:
>
> Multi-teacher KD employs the ensemble of each teacher's prediction output to make more comprehensive and reasonable decision. However, most of previous multi-teacher KD approaches suffers from two issues: 1) diversity losing of the ensemble prediction output and 2) capacity gap between high-capacity teacher ensemble and low-capacity student. To solve the above issues, we propose a novel distillation paradigm named AutoSKD which stochastically samples a teacher from a multi-level teacher team via a learnable categorical distribution, to distill student in a one-to-one manner.
>
> To contextualize the results in this paper, we now summarize our key contributions:
>
> 1. **A novel multi-teacher distillation paradigm named AutoSKD which can adequately utilize the extreme strong teacher and the weak-capacity teachers:** On the one hand, AutoSKD employs the stochastic one-to-one distillation manner to extract more useful knowledge from each teacher (even those teacher whose capacities are weaker than the student) for obtaining high performance. On the other hand, in the multi-level teacher team, those teacher assistants [2] whose capacities are stronger than student model but weaker than the strongest teacher, fill the capacity gap between the strongest teacher and weak student, i.e., AutoSKDBERT can adequately utilize the powerful capacity of strong teacher without the issue of capacity gap [2].
>
> 2. **A two-phase optimization framework for learning the categorical distribution of AutoSKD:** Phase-1 optimization distinguishes effective teachers from ineffective teachers. Phase-2 optimization further optimizes the sampling weights of the effective teachers to obtain satisfactory categorical distribution. The proposed optimization framework avoids employing ineffective teacher in the teacher team to achieve unsatisfactory performance.
>
> 3. **A gradient-based update strategy named SSWO for adapting AutoSKD:** Based on the strategy of continuous relaxation, we develop it to SSWO which is proposed to fill the consistency gap between the categorical distribution optimization and evaluation in terms of the teacher's output.
>
> Moreover, the proposed approach is simple, general and effective so that it is easy to follow by researchers in the future. Taken together, these contributions represent novel insights to the literature on the effect of multi-teacher knowledge distillation.
>
> [1] Hinton G, Vinyals O, Dean J. Distilling the knowledge in a neural network[J]. arXiv preprint arXiv:1503.02531, 2015, 2(7).
>
> [2] Mirzadeh S I, Farajtabar M, Li A, et al. Improved knowledge distillation via teacher assistant[C]//Proceedings of the AAAI conference on artificial intelligence. 2020, 34(04): 5191-5198.
>
> Thank you for the feedback, we hope we have adequately addressed your concerns. We will be happy to answer any additional questions. We encourage you to reconsider your score in light of our updates.

---

### Official Review · Reviewer_jsZh · 2022-11-03

**Confidence:** 4
**Correctness:** 3
**Technical Novelty And Significance:** 3
**Empirical Novelty And Significance:** Not applicable
**Recommendation:** 3

**Clarity, Quality, Novelty And Reproducibility:**

- This paper provides a new paradigm to distill a student model from multiple teacher models. It's mainly about application and techniques. Some critical parameters need manual tuning.
- Some clarification of details is not very clear. But the main idea and motivation are clear.
- This work should be original.


**Strength And Weaknesses:**

### Pros:
1. The proposed method is simple yet effective, and it provided a more flexible way to give full play to the teacher team.
2. The experimental results show obvious superiority compared to the SOTA baselines on the GLUE benchmark.

### Cons:
1. One critical problem is how to choose the ineffective teacher number, i.e., $m$. It is a very important parameter in the whole paradigm. However, as claimed in Sec 3.2.2, $m$ is a hyperparameter and it may differ a lot for different tasks.
2. As claimed in Sec 3.3.2, AutoSKDBERT delivers 25 categorical distribution candidates and selects the best one. How to produce 25 candidates in detail?
3. As stated in Sec 3.3.1 and Sec 4.1, are there 25 categorical distribution candidates in both phase-1 and phase-2? Or phase-1 will only produce 25 candidates in the ablation study experiments?
4. In Sec 3.3.1, what does *trains them from scratch* mean? What is *them*? the student model?
3. In figure 2, we can see $T_{14}$ plays a significant role, but $T_{13}$, which has the same number of parameters as $T_{14}$, is much weaker. The only difference is that $T_{14}$ adopts *whole word masking*. Since *whole word masking* is signally effective, why not adopt it on other teacher models?
4. Because $T_{14}$ is a new type of BERT model with better performance, do the baselines use such a model as one of the teacher models? If not, what are the results without $T_{14}$? What if the baselines use the same teacher models?
6. Since some teacher models, e.g., $T_{14}$, perform distinctly better, it will be good to compare with the best single-teacher KD.
5. What's the number of teacher models of the baselines? Is it a fair comparison if the baselines use fewer teacher models than 14?
8. In Sec 4.2, for the CR setting, the training involves all the teachers, why not use the same way in evaluation? Then it will be consistent between training and evaluation and can compare with SSWO more fairly.
1. It seems unclear about the categorical distribution initialization module. Why is the initialized $\theta_i$ like shown in Figure 1, i.e., why are 0.199, 0.201, 0.199, 0.200, and 0.201?



**Summary Of The Paper:**

The paper proposed a new knowledge distillation paradigm for BERT, AutoSKDBERT, which stochastically sampled a teacher from the predefined teacher team by a categorical distribution. They proposed a two-phase optimization framework to better incorporate the categorical distribution and student model. To alleviate the gap between categorical distribution optimization and evaluation, they proposed an SSWO strategy for optimization. The experimental results show superiority compared with strong baselines.

**Summary Of The Review:**

Overall, the proposed method is simple and easy to implement. And it could raise better performance than SOTA. But there are lots of tricks to boost the performance, like 1. generating 25 candidates, training 25 student models, and choosing the best one; 2. mutual-tuning $m$; 3. using a stronger teacher model (*whole word masking*). However, there lack of details and analysis of the tricks. Further, the clarification should be clearer.

---

> ### Author Response · Authors · 2022-11-19
> **Part 1 of Response to Reviewer jsZh**
>
> We thank you for your comments and feedback. In addition to the general updates, we address your concerns here.
>
> **Q1: One critical problem is how to choose the ineffective teacher number, i.e., m. It is a very important parameter in the whole paradigm. However, as claimed in Sec 3.2.2, m is a hyperparameter and it may differ a lot for different tasks.**
>
> A1:
>
> **How to choose the ineffective teacher number**.
>
> A simple way to choose the ineffective teacher number $m$ is elaborately designing the teacher team and setting $m$ to the number of weak teachers whose capacities are weaker than student.
>
> **How to design appropriate teacher team**.
>
> First, we should determine the strongest teacher and student. Next, we select several teacher assistants whose capacities are stronger than student but weaker than the strongest teacher. Finally, we choose also several weak teachers whose capacities are weaker than student. Above all, the predefined teacher team consists of several weak teachers, several teacher assistants and the strongest teacher.
>
> Taking AutoSKDBERT as an example, we can choose $T_{09}$ to $T_{14}$ as the teacher team which consists of 1 weak teacher (i.e., $T_{09}$ whose capacity is weaker than student) and 5 strong teachers (i.e., $T_{10}$ to $T_{14}$ whose capacities are stronger than student). In this case, we set the ineffective teacher number to 1. Furthermore, we use the above way to design the teacher team of AutoSKD for image classification on CIFAR-100 (see Appendix B in the revised manuscript). Experimental results show that the proposed distillation paradigm can also achieve novel performance when using the simple way to design the teacher team.
>
> **Motivation of using $T_{01}$ to $T_{14}$ as the teacher team in this paper**. The proposed AutoSKD can preserve the diversity of each teacher and fill the capacity gap between strong teacher and weak student. On the one hand, we employ weak $T_{01}$ to $T_{09}$ to verify a guess that the diversities of those weak teachers contribute to improve the distillation performance or not. On the other hand, under a conclusion that the extreme strong teacher (i.e., $T_{13}$ and $T_{14}$) can not always contribute to improving the distillation performance (see Appendix C.4 and G in the revised manuscript), we employ strong $T_{13}$ and $T_{14}$ to verify the effectiveness of the proposed distillation paradigm for capacity gap alleviation.
>
> Moreover, we have added the above contents about ineffective teacher number determination and teacher team design to the revised manuscript (see Appendix A).
>
> **Q2: As claimed in Sec 3.3.2, AutoSKDBERT delivers 25 categorical distribution candidates and selects the best one. How to produce 25 candidates in detail?**
>
> A2: During the optimization process, the current categorical distribution is reported every fixed iteration step. For different tasks, the fixed iteration steps are various. However, AutoSKDBERT generates 50 candidates when entire optimization process completing for each task. For the task of MRPC, RTE and CoLA (training for 50 epochs, i.e., 25 for phase-1 optimization and 25 for phase-2 optimization), the current categorical distribution is reported and treated as a candidate when each epoch completing.
>
> **Q3: As stated in Sec 3.3.1 and Sec 4.1, are there 25 categorical distribution candidates in both phase-1 and phase-2? Or phase-1 will only produce 25 candidates in the ablation study experiments?**
>
> A3: As described in A2, both phase-1 and phase-2 optimization generate 25 candidates. But, those candidates generated in phase-1 optimization are only evaluated in the ablation study experiments.
>
> **Q4: In Sec 3.3.1, what does trains them from scratch mean? What is them? the student model?**
>
> A4: Yes, we train the student model with 25 candidates from scratch to choose the optimal categorical distribution. To improve the readability, we have modified the description in the revised manuscript (see Section 3.3.1).
>
> **Q5: In figure 2, we can see T14 plays a significant role, but T13, which has the same number of parameters as T14, is much weaker. The only difference is that T14 adopts whole word masking. Since whole word masking is signally effective, why not adopt it on other teacher models?**
>
> A5: Whole word masking is used in the process of pre-training which is usually very time-consuming. To avoid training BERT-style teachers from scratch, we directly download the pre-trained teachers from official implementation of BERT, i.e., https://github.com/google-research/bert. Indeed, the performance of AutoSKDBERT can be further improved via whole word masking based teacher team. However, this paper focuses on verifying the effectiveness of the proposed distillation paradigm instead of refreshing the SOTA result.

---

> > ### Author Response · Authors · 2022-11-19
> > **Part 2 of Response to Reviewer jsZh**
> >
> > **Q6: Because T14 is a new type of BERT model with better performance, do the baselines use such a model as one of the teacher models? If not, what are the results without T14? What if the baselines use the same teacher models?**
> >
> > A6:
> >
> > **Do the baselines use such a model as one of the teacher models?**
> >
> > For the comparative methods shown in this paper, $T_{14}$ is only used by MT-BERT [1].
> >
> > **If not, what are the results without T14? What if the baselines use the same teacher models?**
> >
> > Due to the time limitation, we do not show the results of AutoSKDBERT without $T_{14}$.
> >
> > As described in general response, we present the results of 1) multi-teacher AvgKD [2] with $T_{01}$ to $T_{14}$ and $T_{10}$ to $T_{14}$ (see Appendix C.4 in the revised manuscript), 2) multi-teacher TAKD [3] with $T_{01}$ to $T_{14}$ and $T_{10}$ to $T_{14}$ (see Appendix C.4 in the revised manuscript), and 3) TinyBERT [4] with $T_{14}$ (see Appendix G in the revised manuscript). All experimental results show that $T_{14}$ can not always contribute to improving the distillation performance.
> >
> > **Q7: Since some teacher models, e.g., T14, perform distinctly better, it will be good to compare with the best single-teacher KD.**
> >
> > A7: We have show the distillation results of single-teacher KD using $T_{01}$ to $T_{14}$ in the revised manuscript (see Appendix C.4).
> >
> > **Q8: What's the number of teacher models of the baselines? Is it a fair comparison if the baselines use fewer teacher models than 14?**
> >
> > A8:
> >
> > **What's the number of teacher models of the baselines?**
> >
> > Most of the baselines employ single teacher, e.g., TinyBERT [4], MoEBERT [5]. Only MT-BERT [1] employs five teachers (i.e., $T_{10}$ to $T_{14}$) for student distillation.
> >
> > **Is it a fair comparison if the baselines use fewer teacher models than 14?**
> >
> > In Appendix C.4 of the revised manuscript, we discuss the performances of AvgKD and TAKD with various teacher number. Experimental results show that increasing the number of teachers can not always contribute to improving the distillation performance. Moreover, AutoSKDBERT employs $T_{01}$ to $T_{09}$ whose capacities are weaker than the student for knowledge distillation. Above all, the comparison is fair.
> >
> > **Q9: In Sec 4.2, for the CR setting, the training involves all the teachers, why not use the same way in evaluation? Then it will be consistent between training and evaluation and can compare with SSWO more fairly.**
> >
> > In the ablation study experiments, we employ CR and developed SSWO to learn the categorical distribution for AutoSKDBERT. The mechanism of the proposed distillation paradigm is that only a single teacher can be sampled from the teacher team to distill the student. Consequently, we can not use the same way used in CR for evaluation.
> >
> > **Q10: It seems unclear about the categorical distribution initialization module. Why is the initialized θi like shown in Figure 1, i.e., why are 0.199, 0.201, 0.199, 0.200, and 0.201?**
> >
> > A10:  First, we employ torch.randn to generate an initial categorical distribution following N(0, 1). Subsequently, the function of softmax is used to generate the normalized categorical distribution, i.e., 0.199, 0.201, 0.199, 0.200, and 0.201 in Figure 1. The used torch-style code is  "torch.autograd.Variable(1e-3*torch.randn(k).to(self.device), requires_grad=True)".

---

> > > ### Author Response · Authors · 2022-11-25
> > > **Part 3 of Response to Reviewer jsZh**
> > >
> > > **Q11: AutoSKDBERT employs lots of tricks to boost the performance, like 1. generating 25 candidates, training 25 student models, and choosing the best one; 2. mutual-tuning $m$; 3. using a stronger teacher model (whole word masking). However, there lack of details and analysis of the tricks. Further, the clarification should be clearer.**
> > >
> > > A11:
> > >
> > > 1.**Choosing the best one from 25 candidates:** The optimization of categorical distribution is similar to the procedure of architecture search in NAS [6]. Consequently, we follow previous NAS works to choose the best one from multiple architecture candidates.
> > >
> > > Moreover, we have also compared the distillation cost of TinyBERT and our AutoSKDBERT (see Appendix G in the revised manuscript). Experimental results show that the distillation cost of AutoSKDBERT (25 candidates) is $8.4\times$ less than TinyBERT (6 groups of hyper-meters).
> > >
> > > 2.**Mutual-tuning $m$:** The tuning of $m$ can be simple, and we have given some strategies for teacher team design and $m$ determination (see Appendix A in the revised manuscript). In this paper, we employ complex multi-level teacher team motivated by:
> > >
> > > The proposed AutoSKD can preserve the diversity of each teacher and fill the capacity gap between strong teacher and weak student. On the one hand, we employ weak  to  to verify a guess that the diversities of those weak teachers contribute to improve the distillation performance or not. On the other hand, we employ strong $T_{13}$ and T$_{14}$ to verify the effectiveness of the proposed distillation paradigm for capacity gap alleviation.
> > >
> > > 3.**Using a stronger teacher model (whole word masking):** We have drawn a conclusion that the extreme strong teacher (i.e., $T_{13}$ and T$_{14}$) can not always contribute to improving the distillation performance (see Appendix C.4 and G in the revised manuscript).
> > >
> > > [1] Wu C, Wu F, Huang Y. One teacher is enough? pre-trained language model distillation from multiple teachers[J]. arXiv preprint arXiv:2106.01023, 2021.
> > >
> > > [2] Hinton G, Vinyals O, Dean J. Distilling the knowledge in a neural network[J]. arXiv preprint arXiv:1503.02531, 2015, 2(7).
> > >
> > > [3] Mirzadeh S I, Farajtabar M, Li A, et al. Improved knowledge distillation via teacher assistant[C]//Proceedings of the AAAI conference on artificial intelligence. 2020, 34(04): 5191-5198.
> > >
> > > [4] Jiao X, Yin Y, Shang L, et al. Tinybert: Distilling bert for natural language understanding[J]. arXiv preprint arXiv:1909.10351, 2019.
> > >
> > > [5] Zuo S, Zhang Q, Liang C, et al. MoEBERT: from BERT to Mixture-of-Experts via Importance-Guided Adaptation[J]. arXiv preprint arXiv:2204.07675, 2022.
> > >
> > > [6] Pham H, Guan M, Zoph B, et al. Efficient neural architecture search via parameters sharing[C]//International conference on machine learning. PMLR, 2018: 4095-4104.
> > >
> > > Thank you for the feedback, we hope we have adequately addressed your concerns. We will be happy to answer any additional questions. We encourage you to reconsider your score in light of our updates.

---

### Official Review · Reviewer_n9fn · 2022-11-04

**Confidence:** 3
**Correctness:** 4
**Technical Novelty And Significance:** 4
**Empirical Novelty And Significance:** Not applicable
**Recommendation:** 6

**Clarity, Quality, Novelty And Reproducibility:**

Some places of the paper writing is unclear. The proposed method is novel, and the experimental setting looks clear.

**Strength And Weaknesses:**

Strength:
* The authors propose a reasonable bi-level optimization for the weighting of teachers and achieve strong results.

Weakness:
* The paper writing is comparatively weak, e.g., The figure 1 is even harder to understand than the main text part.
* Lack of meaningful analysis and discussion towards the result numbers. For example, overall the proposed approach performs on par with MoBERT. Why in some tasks it performs worse? Is the proposed approach orthogonal to the previous methods? Will the performance be better if multiple techniques are applied together?

**Summary Of The Paper:**

This paper proposes a knowledge distillation approach with multiple teachers by introduce a weighting on teachers, and the weights are optimized in a bi-level manner. It achieves state-of-the-art results on several GLUE tasks compared with other model compression methods.

**Summary Of The Review:**

This paper proposes a reasonable approach to weight multiple teacher in knowledge distillation, and gets good results on the GLUE benchmark, but the results are not discussed and analyzed thoroughly, and the paper writing is comparatively weak.

---

> ### Author Response · Authors · 2022-11-19
> **Response to Reviewer n9fn**
>
> We thank you for your comments and feedback. In addition to the general updates, we address your concerns here.
>
> **Q1: The paper writing is comparatively weak, e.g., The figure 1 is even harder to understand than the main text part.**
>
> A1: We have through revised the manuscript to improve the readability. Particularly, we have added several text descriptions to Figure 1 in the revised manuscript for better expression.
>
> **Q2: Lack of meaningful analysis and discussion towards the result numbers. For example, overall the proposed approach performs on par with MoEBERT. Why in some tasks it performs worse? Is the proposed approach orthogonal to the previous methods? Will the performance be better if multiple techniques are applied together?**
>
> A2:
>
> **Why in some tasks it performs worse?**
>
> Compared to MoEBERT, AutoSKDBERT achieves better performance on 4 out of 7 tasks, in addition to CoLA, QQP and MNLI. There are two main reasons for the above phenomenon: 1) MoEBERT employs more complex student whose architecture is an ensemble of multiple experts, and 2) MoEBERT utilizes extra distillation procedure, i.e., transformer layer distillation. Moreover, the above analysis and discussion have been added to the revised manuscript (see Section 3.4)
>
> **Is the proposed approach orthogonal to the previous methods?**
>
> Yes, the proposed distillation paradigm is orthogonal to most of the previous methods. For example, the transformer layer distillation and data augmentation used in TinyBERT can be combined with our approach. Particularly, each teacher in the teacher team should has same hidden size of transformer layer to the student model for transformer layer distillation.
>
> **Will the performance be better if multiple techniques are applied together?**
>
> We have combined AutoSKDBERT with the transformer layer distillation (TD) and data augmentation (DA) used in TinyBERT to obtain the performance, and shown the results in Table 16 of Appendix D in the revised manuscript.
>
> As shown in Table 16, the combination of AutoSKDBERT and DA shows better performance compared to vanilla AutoSKDBERT on CoLA and QNLI tasks. Furthermore, the combination of AutoSKDBERT, TD and DA achieves better performance compared to vanilla AutoSKDBERT on QNLI tasks. The main reason in our consideration is that the categorical distributions are learned on the vanilla dataset instead  of the augmentation data. In the future, we will directly learn the categorical distribution on the augmentation data.
>
> However, the combination of AutoSKDBERT and TD is prone to obtaining worse performance on each downstream task. We consider that the main cause of the above phenomenon is the knowledge transfer gap between transformer layer distillation and prediction layer distillation, i.e., only using $T_{12}$ for transformer layer distillation, $T_{01}$ to $T_{14}$ for prediction layer distillation. In the future, we will select appropriate teacher team to distill the transformer layer of BERT.
>
> Moreover, the above analysis and discussion have been added to the revised manuscript (see Appendix D).
>
> Thank you for the feedback, we hope we have adequately addressed your concerns. We will be happy to answer any additional questions. We encourage you to reconsider your score in light of our updates.

---

### Author Response · Authors · 2022-11-19
**General Response and Summary of Updates to Manuscript**

We thank to all reviewers for very careful comments.  First, we provide a high-level summary of the changes that we've made to the draft to address reviewers' feedbacks, and conclude with an overview of our key contributions.
The summary of updates that we've made to the draft is given below:
1. We have added a simple guidance for ineffective teacher number determination and teacher team design (see Appendix A in the revised manuscript). (**Reviewer jsZh**)
2. We have added the experiment for image classification on CIFAR-100 (see Appendix B in the revised manuscript), to verify 1) the generality of the proposed distillation paradigm and 2) the simplicity of ineffective teacher number determination. (**Reviewers FHc1 and jsZh**).
3. We have added the distillation results of 1) single-teacher KD, 2) multi-teacher AvgKD [1] and 3) multi-teacher TAKD [2] (see Appendix C.4 in the revised manuscript), to make a fair comparison with AutoSKDBERT and draw a conclusion that extreme strong teacher can not always contribute to improving the distillation performance. (**Reviewers SQHt and jsZh**)
4. We have added more meaningful analysis and discussion for AutoSKDBERT. For instance, 1) we have analyzed the reason of obtaining worse performance on several tasks (see Section 3.4 in the revised manuscript), 2) we have verified that the proposed approach is orthogonal to the previous methods and can achieve better performance on several tasks using the combination of our approach and other techniques (see Appendix D in the revised manuscript). (**Reviewer n9fn**)
5. We have added the experiment of AutoSKDBERT with various hyper-parameters for categorical distribution optimization (see Appendix E in the revised manuscript), to show the impact of each hyper-parameter for distillation performance. (**Reviewer FHc1**)
6. We have added the experiment of distillation cost of AutoSKDBERT, and compared to TinyBERT (see Appendix F in the revised manuscript). (**Reviewers FHc1 and jsZh**)
7. We have added the experiment of TinyBERT with the strongest teacher $T_{14}$ (see Appendix G in the revised manuscript), to examine that the stronger teacher can not always contribute to improving the distillation performance due to the capacity gap [2] between teacher and student. (**Reviewer jsZh**)

To end this update, we discuss a common concern across all reviewers.

**Fair Comparison**

In this paper, we employ a multi-level teacher team to distill AutoSKDBERT. The multi-level teacher team consists of 14 teachers: 9 teachers weaker than student and 5 teachers stronger than student. Moreover, 2 out of 5 strong teachers are stronger than the teacher used in previous works (i.e., $T_{12}$ in this paper). Above all, in AutoSKDBERT, the number of the teachers is more, and the capacity of the best teacher is stronger. However:

1. **Increasing the number of teachers can not always improve the distillation performance**: In Appendix C.4 of the revised manuscript, we discuss the performances of AvgKD and TAKD with various teacher number. Experimental results show that increasing the number of teachers can not always contribute to improving the distillation performance.
2. **Employing stronger teacher can not always improve the distillation performance**: In Appendix G of the revised manuscript, we use the strongest teacher used in AutoSKDBERT, i.e. $T_{14}$, to distill previous SOTA TinyBERT. Experimental results show that $T_{14}$ based TinyBERT achieves worse performance on 4 out of 5 tasks than $T_{12}$ based one, due to the capacity gap between $T_{14}$ and student. Moreover, the results of single-teacher KD in Table 15 (see Appendix C.4 in the revised manuscript) show similar phenomenon. As a result, we can draw a conclusion that employing stronger teacher can not always improve the distillation performance too.
3. **AutoSKDBERT can adequately utilize the extreme strong teacher and the weak-capacity teachers**. On the one hand, AutoSKDBERT employs several teacher assistant [2] in multi-level teacher team to fill the capacity gap between the extreme strong teacher and the weak student. On the other hand, AutoSKDBERT adequately utilize the useful knowledge of single teacher, even the weak-capacity teacher whose capacity is weaker than the student.

[1] Hinton G, Vinyals O, Dean J. Distilling the knowledge in a neural network[J]. arXiv preprint arXiv:1503.02531, 2015, 2(7).

[2] Mirzadeh S I, Farajtabar M, Li A, et al. Improved knowledge distillation via teacher assistant[C]//Proceedings of the AAAI conference on artificial intelligence. 2020, 34(04): 5191-5198.

---

### Decision · Program_Chairs · 2023-01-20

**Decision:**

Reject

**Justification For Why Not Higher Score:**

* Lack of sufficient experimental comparisons and analyses
* Writing needs to be improved

**Justification For Why Not Lower Score:**

N/A

**Metareview: Summary, Strengths And Weaknesses:**

The paper proposes a new knowledge distillation approach that uses multiple teacher models with weights. The teacher weights are optimized in a bi-level manner. The approach shows improved performance on GLUE benchmark compared to certain baseline methods. The reviewers have found that more experimental comparisons and analyses are needed to validate the effectiveness of the approach, such as whether the improvement is due to the new knowledge distillation approach per se or the use of multiple teachers, and others. The paper presentation also needs to improve.